# Sampling bias overestimates climate change impacts on forest growth in the southwestern United States

Stefan Klesse [1], R. Justin DeRose[2], Christopher H. Guiterman[1], Ann M. Lynch[1,3], Christopher D. O'Connor[4], John D. Shaw[2] & Margaret E.K. Evans[1]

Climate—tree growth relationships recorded in annual growth rings have recently been the basis for projecting climate change impacts on forests. However, most trees and sample sites represented in the International Tree-Ring Data Bank (ITRDB) were chosen to maximize climate signal and are characterized by marginal growing conditions not representative of the larger forest ecosystem. We evaluate the magnitude of this potential bias using a spatially unbiased tree-ring network collected by the USFS Forest Inventory and Analysis (FIA) program. We show that U.S. Southwest ITRDB samples overestimate regional forest climate sensitivity by 41–59%, because ITRDB trees were sampled at warmer and drier locations, both at the macro- and micro-site scale, and are systematically older compared to the FIA collection. Although there are uncertainties associated with our statistical approach, projection based on representative FIA samples suggests 29% less of a climate change-induced growth decrease compared to projection based on climate-sensitive ITRDB samples.

[1] Laboratory of Tree-Ring Research, University of Arizona, 1215 East Lowell Street, Tucson, AZ 85721, USA. [2] U.S. Forest Service, Rocky Mountain Research Station, Forest Inventory and Analysis, 507 25th Street, Ogden, UT 84401, USA. [3] U.S. Forest Service, Rocky Mountain Research Station, 1215 East Lowell Street, Tucson, AZ 85721, USA. [4] U.S. Forest Service, Rocky Mountain Research Station, 800 East Beckwith Avenue, Missoula, MT 59801, USA. Correspondence and requests for materials should be addressed to S.K. (email: sklesse@gmx.net)

Projected increases in global temperatures for the twenty-first century exceed the variability of the past several centuries[1]. Semi-arid forests, in particular, have been identified as vulnerable to global warming[2,3]. Rising temperatures and associated increasing evaporative demand[4] will increase the frequency, intensity, and duration of droughts in many semi-arid regions. Resulting declines in tree growth and vigor[5,6] potentially increase tree vulnerability to insect attack, contributing to large-scale outbreaks that amplify the magnitude and scale of mortality events[7]. Warmer and drier conditions, combined with a history of fire suppression, have increased wildfire activity[8] and the potential for high-severity fire[9]. Cumulatively, these effects have the potential to reduce the carbon sink strength of semi-arid forests, along with other ecosystem services that they provide. Although increasing forest vulnerability with sustained global warming is generally supported, particularly in water-limited areas, there are few direct sources of information to quantify the magnitude of climate change impacts.

Tree-ring networks provide rich information on tree growth response to environmental variation at regional to hemispheric scales[10,11] and have recently been used to forecast future forest growth. Growth reductions of up to 70% have been projected for forests of the interior western U.S. in the second half of the twenty-first century compared to the first half of the twentieth century[12]. Across the southwestern U.S., a tree-ring-derived "forest drought stress index", which correlates well with tree mortality, bark-beetle outbreaks, fire, and remotely sensed productivity, has been projected to be more severe by the 2050s than the most severe 6-year drought conditions of the last 1000 years[5].

However, it is recognized that climate sensitivity estimated from tree-ring records in the International Tree-Ring Data Bank (ITRDB) may overestimate forest vulnerability to climate change[12,13], because many of the samples in the ITRDB are the result of targeted sampling of old trees on ecologically marginal sites in order to maximize climatic signal in ring-width variation (c.f. "The site and tree selection principles of dendrochronology", Fig. 1)[14]. In fact, the ecological and geographic distributions of individual tree species encompass a range of climatic and edaphic conditions (e.g., deeper soils, cooler aspects) across which climate may be a more or less important factor limiting tree growth[14–16]. Representative sampling across these gradients is necessary to characterize the response of a forest ecosystem to climate variability.

In this study we analyze and quantify the consequences of targeted sampling of old trees on marginal sites by comparing ITRDB chronologies from the U.S. Interior West against systematically sampled tree-ring collections. Currently the most extensive systematically sampled tree-ring collection in the region, the U.S. Forest Service Interior West-Forest Inventory and Analysis (FIA)[17] dataset includes 4655 trees from permanent forest inventory plots. The FIA plot network is designed to representatively sample all forested lands of the U.S., with one plot per 2428 ha, resulting in a dataset that is, to a large degree, spatially and ecologically unbiased[18]. We augment the FIA sample with two densely sampled, landscape-scale collections including 828 trees in Arizona and New Mexico[19,20] (Fig. 2, cf. Methods section "Tree-ring data") and refer to this combined dataset as the FIA or "inventory" collection.

We analyze ring-width time series from forest inventory vs. targeted (ITRDB) samples, evaluating the expectation that both ring-width variability and climate sensitivity are higher in the targeted sample. Our analyses focus on some of the most widespread, important, and densely sampled tree species in the Interior West—Douglas-fir (Pseudotsuga menziesii var. glauca, PSME), ponderosa pine (Pinus ponderosa, PIPO) and common pinyon (Pinus edulis, PIED). We investigate possible explanations for differences in year-to-year ring-width variability between the two samples, with age, macro-site, and micro-site selection as alternative (but not mutually exclusive) hypotheses (Fig. 1). Finally, we demonstrate how much future tree growth projections across the southwestern U.S. differ when based upon targeted, old trees vs. the representative, inventory samples. That is, we address the following three questions: (1) Do targeted vs. forest inventory tree-ring collections differ in growth variability and climate sensitivity? (2) What factors explain the differences? (3) What are the

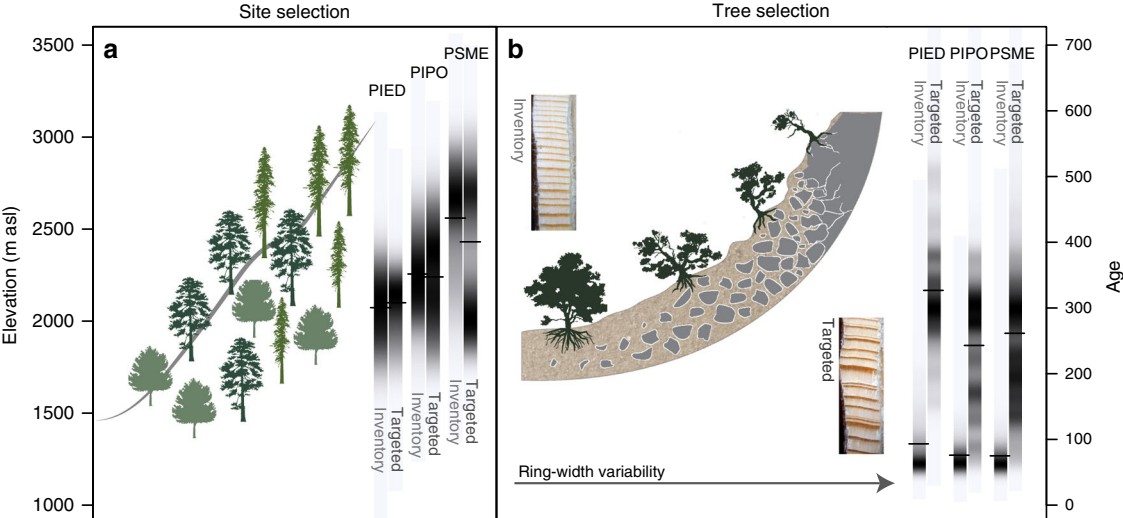

**Fig. 1** The site and tree selection principles of dendrochronology. The majority of tree-ring time series in the International Tree-Ring Data Bank (ITRDB) were sampled to maximize chronology length and climate sensitivity. **a** Comparison of the elevation at which three important species (PIED = *Pinus edulis*, PIPO = *Pinus ponderosa*, PSME = *Pseudotsuga menziesii*) were sampled in Arizona, Colorado, New Mexico, and Utah. A bias towards sampling low-elevation sites (therefore, warmer and drier) is evident for ITRDB samples of PSME compared to the representative inventory (Forest Inventory and Analysis; FIA) dataset, leading to a "macro-site selection bias". **b** Selection of trees on steep, rocky slopes with little soil water holding capacity leads to greater ring-width variability (modified from Fritts[14]). We hypothesize that this sampling practice—the "micro-site selection bias"—explains higher standard deviation of ring-width index in the ITRDB collection (Fig. 2), even after accounting for elevation, age, and other factors. Vertical bars show the age distribution of the two collections in 1995. Tick marks indicate the median in each density strip

implications for projection of future forest growth in response to climate-related stress?

## Results

**Growth variability and climate sensitivity.** We assessed the growth variability of trees at each ITRDB sampling site over the 1930–1995 period, measured as the standard deviation (SD) of detrended ring-width time series, and compared this against growth variability (SD) of all FIA tree-ring time series of the same species within a search radius of 100 km. Less variability indicates more stable growth—i.e., a weaker response to temporally variable environmental conditions (macroclimatic variation)[21].

Throughout the interior west, targeted samples of all three species show higher growth variability compared to nearby forest inventory samples (Fig. 3a). Mean growth variability (SD) of Douglas-fir is 9 ± 20% (mean ± standard deviation) higher in the ITRDB collection than in the FIA collection, with no evident

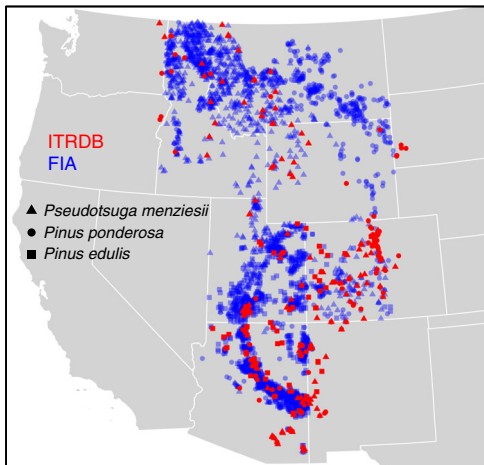

**Fig. 2** Geographic distribution of targeted samples (ITRDB; red) vs. representative forest inventory samples (FIA; blue)

geographic trend. Mean growth variability of common pinyon and ponderosa pine were 11 ± 18% and 14 ± 26% higher, respectively, in the ITRDB sample, and this difference was significantly greater in the south (interaction effect between database and latitude; $p < 0.001$, $n = 2648$ and 4178, respectively). In addition, growth variability increased with decreasing elevation, from north to south, and west to east, i.e., all in the direction of increasing aridity[15], as well as with increasing tree age (Supplementary Figure 1). Common pinyon, which occupies a smaller geographic area and ecological niche than the other two species, showed no influence of latitude on growth variability when sampled on forest inventory plots.

We then compared climate sensitivity of the ITRDB vs. FIA collections, i.e., partial regression coefficients from multiple linear regressions explaining tree-level relative ring-width variation as a function of climate data. We used climate variables known to explain ring-width variation in the southwestern United States: mean monthly maximum temperature of the antecedent fall (August to October), current summer temperature (May to July), and cool-season precipitation (November to March)[5,13,14,16].

Ring-width variation across the interior western U.S. is better explained by and is more responsive to climate variation in the great majority of targeted samples compared to nearby forest inventory samples (larger coefficient of determination and steeper regression slopes; Fig. 4). ITRDB ponderosa pine and common pinyon are significantly more negatively sensitive to antecedent fall temperatures compared to nearby forest inventory samples (average difference between regression coefficients: −0.02, $p < 0.001$, $n = 88$ and 63, respectively; Fig. 4a), whereas Douglas-fir shows predominantly positive differences north of 42°N (approximately the southern border of Montana, Supplementary Figure 2a). Targeted samples of Douglas-fir are also less negatively sensitive to summer temperatures compared to forest inventory samples (Supplementary Figure 3). In fact, in the northern Rockies (north of 42°N) they tend to be much more positively sensitive to temperature (Fig. 4c and Supplementary Figure 2b)—which may reflect sampling aimed at snowpack reconstruction[22]—in spite of the fact that the two samples do not differ in elevation in this region ($p = 0.59$, $n = 30$). Targeted

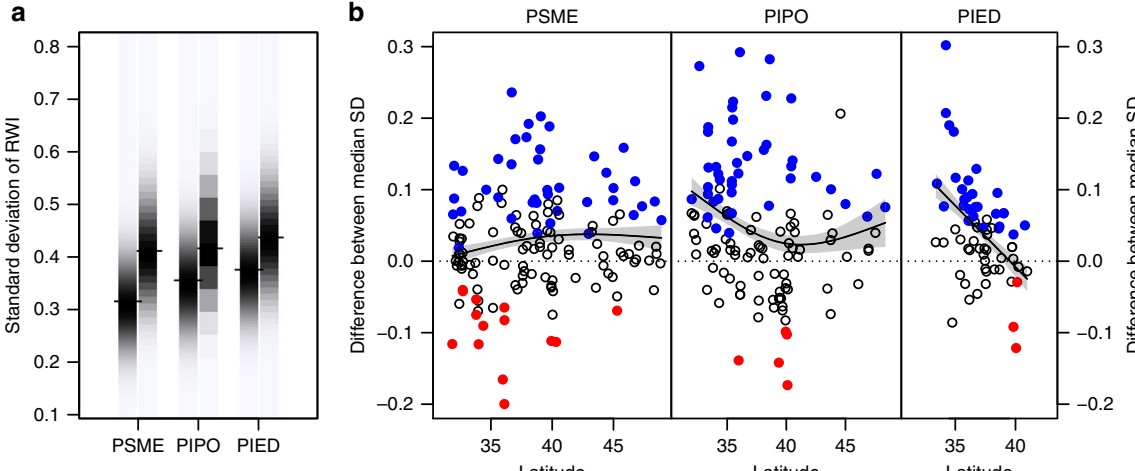

**Fig. 3** Comparisons of growth variability in ITRDB vs. FIA (targeted vs. representative forest inventory) time series. **a** Distribution of the standard deviation (SD) of ring-width index (RWI) time series of the three focal species, with FIA vs. ITRDB data in left vs. right density strips, respectively. Species abbreviations (PSME, PIPO, and PIED) follow Fig. 1. **b** Difference between the median SD of ITRDB vs. surrounding FIA time series (ITRDB median SD minus FIA median SD). Contrasts are filled blue if the median ITRDB SD is significantly greater than the median FIA SD; red if the difference is significantly negative ($p < 0.01$, two-sided Wilcoxon test). The smoothed regression line is the expected difference between the median SD of the two samples with respect to latitude, with 95% confidence intervals

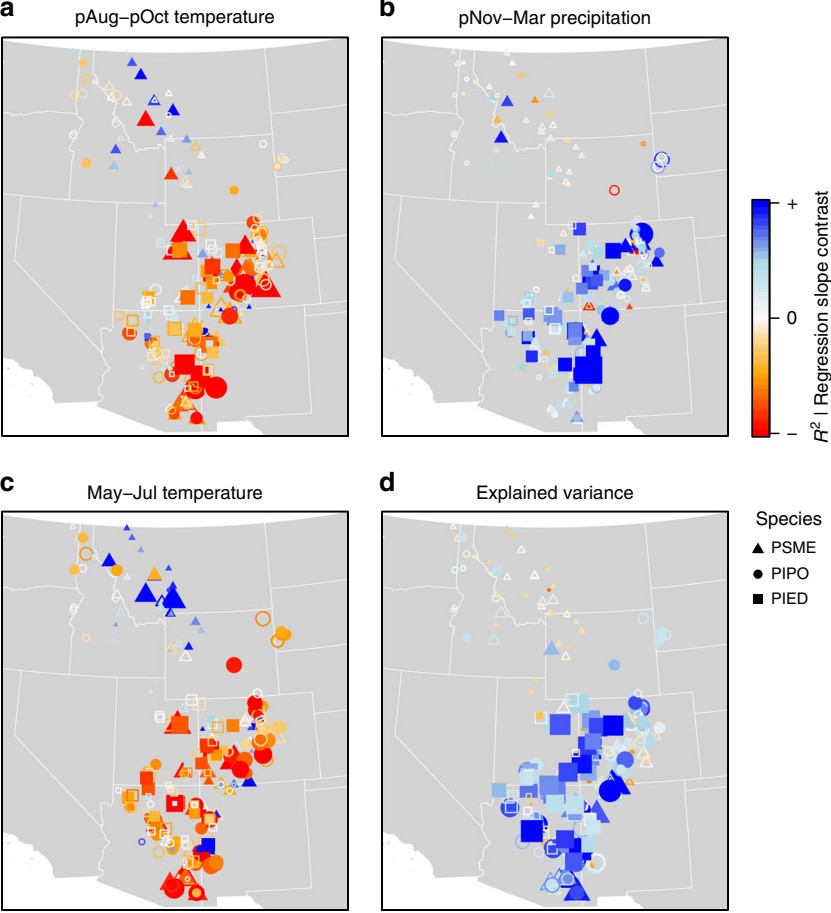

**Fig. 4** Comparison of median climate sensitivity and explained variance of targeted (ITRDB) vs. forest inventory (FIA) samples. Symbol size indicates the absolute magnitude of **a−c** sensitivity to three climate variables (regression slopes) and **d** explained variance ($R^2$) of ITRDB samples, with positive contrasts (ITRDB minus FIA) in blue, and negative contrasts in orange to red. Filled symbols denote significant differences at $p < 0.01$ (two-sided Wilcoxon test) of the FIA distribution surrounding each ITRDB site. $R^2$ contrast values range from −0.30 to +0.41, while slope contrasts range from −0.18 to +0.23 (temperature) and −0.03 to +0.06 (precipitation)

samples of ponderosa pine are generally more negatively sensitive to summer temperatures. Common pinyon and ponderosa pine samples in the ITRDB are, on average, more than twice as sensitive to winter precipitation ($p < 0.001$, $n = 63$ and 88, respectively), with the strongest contrasts in Colorado (Fig. 4b). ITRDB Douglas-fir are on average 17% more sensitive ($p < 0.05$, $n = 128$) with no clear geographical pattern.

Considering just the forest inventory sample, all three species are similarly sensitive to warm season maximum temperatures. We found a stronger negative impact of antecedent fall temperature southward and eastward for Douglas-fir and ponderosa pine ($p < 0.001$, $n = 6127$ and 4178, respectively) and stronger negative impacts of summer temperature on these two species eastward (Fig. 4c). All three species are strongly positively sensitive to winter precipitation, with a significant trend towards greater sensitivity in the south and east. Older trees are significantly more sensitive to all three climate parameters (Supplementary Figure 3).

Climate explains on average 12% more variance in the targeted samples of common pinyon and 10% for ponderosa pine (both $p < 0.001$, $n = 2648$ and 4178, respectively) compared to forest inventory samples. This metric of climate sensitivity, explained variance, increases significantly towards the south for ITRDB samples of Douglas-fir and ponderosa pine ($p < 0.001$, $n = 6127$ and 4178, respectively, Fig. 4d). The geographic trend in $R^2$ was

weak but significant in the FIA sample, resulting in a strong gradient in the contrast of explained variance for Douglas-fir and ponderosa pine with decreasing latitude (Fig. 4d and Supplementary Figure 2d).

**Projection of future growth for the U.S. Southwest**. Differences in climate sensitivity between the targeted vs. representative samples are expected to yield different projections of future tree growth in response to changing climate. To illustrate this point, we used the simple multiple regression model above to project tree growth forward. We aggregated all time series south of 38°N (the southern borders of Colorado and Utah) into two regional chronologies (one ITRDB and one FIA), as in Williams et al.[5]. The two chronologies are strongly correlated ($r = 0.93$, 1902–2008, Fig. 5), with comparable 1- and 2-year lag effects (1-year autocorrelation of 0.368 vs. 0.408 in the ITRDB vs. FIA time series, and 2-year partial autocorrelation of 0.270 vs. 0.241, respectively). However, the chronology derived from targeted sampling has more pronounced variability (SD of 0.29) compared to the chronology derived from forest inventory sampling (SD of 0.20). Strikingly, this difference in SD translates to approximately 50% greater climate sensitivity of the ITRDB chronology compared to the FIA chronology; regression slopes against cool season precipitation and warm season maximum temperatures were 59 and 41%

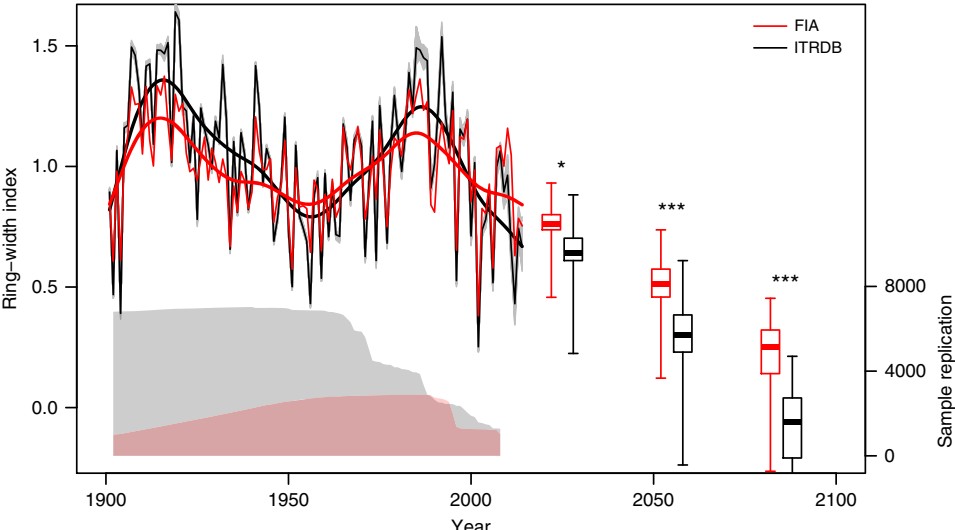

**Fig. 5** Regional ring-width index (RWI) chronologies for the U.S. Southwest (all samples south of 38°N) based on the forest inventory sample (FIA, in red) vs. targeted sample (ITRDB, in black). The gray envelopes indicate variability among 100 iterations of randomly resampling available ITRDB ring-width time series to match the sample replication of the FIA chronology. Boxplots show projected mean growth for 2010–2039, 2040–2069, and 2070–2099 under the "business-as-usual" scenario RCP8.5 across 15 different Atmosphere-Ocean Global Circulation Models (GCM) from the CMIP5 ensemble. Bold horizontal lines denote the median of the projections, the boxes denote the interquartile range, and whiskers extend to the most extreme future RWI values arising from GCM-projected future climate. Asterisks show the significance level of differences in projected future tree growth based upon the two datasets (two-sided $t$ test; *$p < 0.05$; ***$p < 0.001$). Sample replication, i.e., the total number of ring-width time series, is indicated as counts (right-hand $y$ axis) in gray (ITRDB) and red (FIA) shaded areas below the chronologies (right-hand $y$ axis)

steeper ($p < 0.001$ and $p = 0.035$, $n = 214$), respectively:

$$\text{(ITRDB) ring width index} = 0.161 \times \ln(\text{precip}) - 0.139 \times \text{tmax};$$
$$R^2_{\text{adj}} = 0.647,$$
(1)

$$\text{(FIA) ring width index} = 0.101 \times \ln(\text{precip}) - 0.098 \times \text{tmax};$$
$$R^2_{\text{adj}} = 0.607.$$
(2)

That is, the difference in sensitivity to cool season precipitation between ITRDB and FIA chronologies is 0.06 (0.161 minus 0.101), which is 59% of 0.101. Although the average correlation between tree-level time series differs substantially between the ITRDB and FIA samples (rbar=0.328 vs. 0.202, respectively), climate predictors explain nearly the same amount of variance in the two regional chronologies $\left(R^2_{\text{adj}} = 0.647 \text{ vs.} 0.607\right)$.

We used CMIP5 projections of future decadal averages of these two climate variables to project relative radial growth across the U.S. Southwest. A strong decline is projected based upon either tree-ring sample throughout the remainder of the twenty-first century. However, the magnitude of the projected growth decline differs substantially: by the end of the twenty-first century, the median of 15 different general circulation models projects an average growth rate decline of forest inventory trees of about 75% compared to the twentieth century mean, whereas the median projection for the targeted trees is below 0% (106% decrease). Thus, the projection based on forest inventory samples shows 29% less of a growth decline compared to the projection based on targeted samples (106−75 = 31; 31/106 = 29). Equivalently, we find a 41% stronger growth reduction when projection is based on the climate-sensitive trees of the ITRDB compared to the representative forest inventory sample (31/75 = 41; Fig. 5).

Replacing precipitation and mean maximum temperature with climatic water balance and vapor pressure deficit as predictors of tree growth increases the projected relative difference in growth decrease by just 1% (Supplementary Figure 4).

## Discussion
Large-scale inference based upon tree-ring time series sampled from old, targeted trees overestimates the impact of climate change, especially on southwestern U.S. forest growth. This is not a surprise—it has been known for half a century that trees at the arid edge of the forest biome are more sensitive to climate variation—i.e., their growth correlates more strongly with climate and varies from year to year more strongly[15,16]. This observation, made by one of the founding fathers of dendrochronology (H.C. Fritts), has been the basis for targeted sampling by dendroclimatologists for decades. That our tree-by-tree analysis revealed significant differences between ITRDB and FIA samples in the variance explained by climatic parameters verifies that the practice of targeting open-grown trees occurring under ecologically marginal conditions, in order to maximize climate signal with a few samples, achieved that goal.

We found evidence in support of three hypotheses for the observed differences in growth variability and climate sensitivity between the FIA and ITRDB collections: First, Douglas-fir ITRDB sites in the U.S. Southwest are located at noticeably lower elevation than the representative FIA collection (Fig. 1a), reflecting a tendency to target sites that are marginal with respect to its elevation distribution, which we term the "macro-site selection bias". Second, across all three species, there is a striking age difference between the two samples, with the average cumulative ring count of FIA time series 77 years compared to 274 years for ITRDB time series (Fig. 1b). Age is a significant predictor of radial growth variability and climate sensitivity (Supplementary Figure 1 and 3). Age- or size-related differences in climate sensitivity have been reported in several studies, and have been attributed, among other factors, to differences in water and nutrient translocation mechanisms between young (small) and old (large)

individuals[23,24]. Third, in the case of ponderosa pine and common pinyon, growth variability (SD) of ITRDB time series is significantly greater than growth variability of the forest inventory time series, even after controlling for elevation, age, latitude, longitude, climate normals, and two-way interactions between these factors (Supplementary Figure 1). We hypothesize this is caused by targeted sampling of trees found on steep, rocky slopes with less soil water capacity (as illustrated in Fig. 1b, and Fritts et al.[15], Fig. 4), which we refer to as the "micro-site selection bias". Indeed, dendrochronologists have long paid attention to fine-scale site characteristics like landscape position, aspect, slope, and soil depth that affect ecohydrological and micro-meteorological characteristics (runoff, radiation, snow responses, wind, and humidity), driving inter-annual variability in ring widths, which are not captured by coarse-scale climate predictors.

There was clear agreement between the FIA and ITRDB chronologies that the expected net effect of climate change on tree growth is strongly negative (Fig. 5). But by quantifying the bias of the ITRDB towards climate-sensitive trees, we can put projected forest growth decrease based upon ITRDB samples[12,13] in context —an unbiased context that suggests a considerable upward correction (29% in our model) for southwestern U.S. forests. The marginal conditions under which ITRDB samples have been collected represent a very small fraction of the distribution of forest conditions across the U.S. Southwest. The projection of negative growth in the future for this subset of trees suggests that mortality may occur at range margins, consistent with the prediction of range retractions made by climate envelope or species distribution models[25]. However, heterogeneity in the strength of tree growth response to climate variation, and how that heterogeneity is arranged on landscapes, suggests reduced landscape-scale vulnerability of tree growth to drought stress. Decreased growth variability, in turn, has been linked to lower risk of tree mortality[26,27], implying that a representative sample of southwestern U.S. trees may be less vulnerable to mortality than trees growing at the arid edge of their distribution.

Though we have been able to compare projections of future US Southwestern tree growth with respect to the climate-sensitive nature of the ITRDB sample, there remain important caveats associated with our analysis. The first analytical step—detrending—is intended to remove the long-term trend of decreasing ring-width with increasing bole diameter that is imposed by geometry. However, this procedure also removes environmental signals that vary on time scales longer than decadal to multi-decadal, including climate trends, possible increasing water use efficiency associated with $CO_2$ fertilization, other plastic physiological responses, and stand dynamics. Because we neither model these drivers explicitly nor model absolute growth rates, we cannot parse their effects or investigate how they will interact to determine future tree growth. Further, our model only accounts for linear responses of tree growth to climate, i.e., it assumes stationarity of responses, when it is likely that trees' climate-growth responses will change as they experience increasingly unusual climatic conditions, changing phenology, and extreme events. True growth responses to climate are also likely to deviate from our statistical predictions in the face of no-analog combinations of climate variables, including beyond temperature and precipitation, novel covariance structure of the array of plant-relevant environmental variables that influence evapotranspiration. For all these reasons, we draw attention to the difference between projections, caused by biased sampling (poor representation of elevation, age structure, and micro-site conditions). Finally, we point out that our projection does not account for increased vulnerability to insect attack and increased risk of fire, which both have the potential to strongly and suddenly change the carbon source vs. sink balance of forests.

A remaining question is whether a similar systematic climate sensitivity bias exists in tree-ring networks in other parts of the world and, if so, at what magnitude. On the one hand, the potential for bias may be greatest at the arid edge of the forest biome, and the problem may be limited to certain places, like the southwestern United States. Supporting this, Klesse et al.[28] found no significant difference, between targeted (ITRDB) and non-targeted samples, in the sensitivity of tree growth to May—August temperatures in more mesic forests of Central and Northern Europe. On the other hand, we see an indication of positive summer temperature sensitivity bias in ITRDB Douglas-fir samples in the northern Rockies, so there seems to be the potential for other kinds of bias in other regions. Absent information regarding the original purpose of the sampling, we advise caution when using ITRDB chronologies. For example, the environmental space sampled by ITRDB collections (e.g., mean annual temperature and precipitation, elevation) should be compared against the distribution of environmental conditions of the forested study area of interest, to evaluate how representative (at the macro-site scale) those sites might be—keeping in mind that micro-site and age biases are likely to be present as well.

The ITRDB is a treasure trove for records of tree growth variability, both in space and time. However, in order to answer broad-scale ecological questions, and improve our understanding of forest vulnerability to climate change across forest biomes, dendrochronological sampling designs must be more representative—not just within a site[29], but also spatially by covering an ecological gradient[30,31] or with a systematic grid[20,32]. We are optimistic that the establishment of a new data standard on the ITRDB (TRiDaS[33]) that allows additional information about sites (slope, aspect, stand density, soil depth, etc.) and trees (diameter, height, pith offset, etc.) to be readily shared will improve the value and versatility of publicly available data for ecological forecasts of forested ecosystems. Such metadata will enable explicit modeling of the many influences on tree growth, help to close the scaling gap between gridded climate products and the growth conditions experienced by individual trees, and ultimately lead to better estimates of the sensitivity of forest ecosystem productivity to changes in climate.

## Methods

**Tree-ring data.** We focused our analysis on the three species of the interior western U.S. most well-represented in publicly available tree-ring archives (the International Tree-Ring Data Base, ITRDB): *Pseudotsuga menziesii* var. *glauca* (Douglas-fir), *Pinus ponderosa* (ponderosa pine), and *Pinus edulis* (common pinyon). Our data consisted of two independent collections: (1) tree-ring time series downloaded from the ITRDB as of December 2016, and (2) tree-ring time series developed as part of the Interior West Forest Inventory and Analysis (FIA) Program, sampled in a gridded fashion over all forested areas[17]. The latter collection was complemented by two additional inventory-style datasets from (a) the Pinaleño Mountains in southeastern Arizona, where sampling was performed in a systematic grid of 54 0.05-ha circular plots spaced 1 km apart[34], and from (b) northeastern Arizona and northwestern New Mexico, where a subset of forest inventory plots on the Navajo Nation was sampled along a climate gradient[19].

**Interior West-FIA tree-ring data.** The Interior West Forest Inventory and Analysis (IW-FIA) Program encompasses the states of Arizona, Colorado, Idaho, Montana, Nevada, New Mexico, Utah, and Wyoming, U.S.A. Tree-ring data used in this study originated from two inventory designs, periodic (late-1980s – 2001), in which a state was typically inventoried over a period of a few years and not revisited again for a decade or more, and annual (2000 to present), in which the systematic grid of plots is divided into equal, interpenetrating panels of plots that are measured on a continuous 10-year rotation in the interior western states. While periodic FIA inventories covered all forest types in the Interior West at high spatial density (i.e., generally no spatial bias), they were not temporally balanced. Different parts of a given state were measured over the course of a periodic inventory, and different states were inventoried at different times. The current and ongoing FIA annual design is also geographically unbiased, but the interpenetrating panel design simultaneously creates a temporally unbiased sample[35] across the United States. This sampling approach is designed to infer population-level estimates of the Nation's forests[36] through the use of statistical estimators (e.g., forest area, tree volume, or trees per area) applied to plot-level representation, which is approximately one plot per 2428 ha[18]. As part of the IW-FIA data collection program, a single increment core per tree was collected from a subset of trees on plots for the purpose of determining stand age, periodic growth rates, and potential productivity (see ref. [17] for details). Cored trees represented the dominant species on the plot in terms of size (i.e., diameter) and forest type at each location. The number of trees

cored per plot across the Interior West varied from one to as many as 14. For the three species in this study, the average number of cored trees per plot was 1.4. While the tree-ring data in this study have extraordinary spatial representation, our analysis is limited to series that were crossdatable (i.e., some samples were either too complacent or had too many missing rings for year assignments to be made with confidence), and this introduces some (but only minor) filtering of the original sample. We did not include cores from New Mexico in our analysis, because the processing (crossdating and measuring) of those samples is ongoing and cannot yet be considered representative. In total, we included 1670 Douglas-fir samples from 1239 plots, 1830 ponderosa pine samples from 1156 plots, and 717 common pinyon samples from 603 plots.

**Tree-ring data from the Pinaleño Mountains**. The Pinaleño Mountains in southeastern Arizona are the tallest of the Madrean Sky Island ranges, spanning a vertical gradient of more than 2100 m, from Chihuahuan mixed-desert shrubland at 1150 m to spruce-fir forest up to 3268 m. Tree demographic and fire history information across the landscape was collected on a systematic grid of 54 0.05-ha circular plots spaced 1 km apart, established between 2006 and 2009 (c.f. Figure 2 in ref. [20]). Increment cores were collected from all trees with diameter at breast height (DBH) ≥ 19.5 cm throughout each plot, and from trees between one and 19.4 cm DBH on a nested sub-plot equal to one third the area of the full plot (0.017 ha). More detailed information on sampling is found in ref. [34]. We included 43 ponderosa pine samples from 12 plots, and 316 Douglas-fir samples from 43 plots from this study.

**Tree-ring data from the Navajo Nation**. Tree-ring collections from the Navajo Nation in northeastern Arizona and northwestern New Mexico were sampled within a systematic grid of 272 continuous forest inventory (CFI) plots installed between 1974 and 2005 monitored jointly by the Bureau of Indian Affairs and Navajo Forestry Department. This plot grid encompasses ~250,000 ha of largely ponderosa pine-dominated forest, with areas of piñon-juniper and dry mixed-conifer. Each CFI plot consists of three 0.1 ha circular subplots. As described in ref. [19], our subsample includes 36 plots randomly selected across a climate gradient consisting of topographic relative moisture index[37] and elevation. At each plot, 5–15 trees of one to three target species (ponderosa pine, Douglas-fir, or common pinyon) were cored perpendicular to slope at 20–50 cm above ground level. We included 469 trees (908 samples) from 33 plots with ponderosa pine, 13 plots with Douglas-fir, and 11 plots with common pinyon.

**Sample processing**. All increment cores and cross-sections were mounted and surfaced following standard procedures[38]. Samples were crossdated, ensuring the accuracy of years assigned to annual rings, using a combination of visual pattern matching, skeleton plots, and statistical verification using the program COFE-CHA[39] on ring widths measured on a Velmex TA system (0.001 mm precision) and recorded in the software program Measure J2X or Tellervo[40].

**Climate data**. We used climate data from ClimateNA v5.40 [41] (http://tinyurl.com/ClimateNA), which are based on CRU TS3.22 [42] gridded historical monthly data, downscaled and interpolated to adjust the mean and variance of the original CRU time series to elevation- and location-specific values. Future climate projections were taken from 15 Atmosphere-Ocean General Circulation Models of the CMIP5 multimodel dataset, i.e., the IPCC Assessment Report 5 (2013). We focus on climate normals for the 2010–2039, 2040–2069, and 2070–2099 periods under emission scenario RCP8.5, an extra 8.5 W m$^{-2}$ of energy retained by the atmosphere compared to the pre-industrial baseline.

**Statistical analysis**. Because tree size and forest stand information is not available on the ITRDB, we were constrained to analyze tree growth in relative terms. That is, raw ring widths were detrended prior to analysis, controlling for the ontogenetic trend of decreasing radial growth with increasing bole diameter, as well as possible effects of stand dynamics. The dependent variable is thus tree growth in a given year relative to an expected value, following one of several possible detrending methods. Results presented in the main manuscript are based on ratio-detrended ring-width time series using a modified negative exponential curve (substituted by straight line if a curve yielded a poor fit to raw data). Results from two alternative methods of detrending, i.e., spline-detrending with a 50% frequency-cutoff at 30 and 100 years—with and without prewhitening, i.e., the removal of temporal autocorrelation by fitting an autoregressive (AR) or autoregressive moving average (ARMA) model to the original time series—can be found in the supplementary material (Supplementary Figures 1, 5–7). Conclusions were robust to these alternative detrending choices.

We refrained from performing analyses at the site level, because the number of trees sampled per FIA plot is low (usually 1–2). Instead, we took a tree-level approach, comparing geographically specific distributions of values[43]. We used 1930–1995 as the period of analysis, because 1995 is the most common last year in the FIA core collection, at which time median FIA tree age is ~80 years. Ring-width time series that had less than 30 years of overlap with this window were excluded from analysis. Standard deviation was calculated for each detrended time series. Using multiple linear regression, ring-width index of each time series was predicted

as a function of total cool season precipitation (previous November to current year March), mean maximum temperatures in fall of the previous year (August to October), and summer of the current year (May to July) to obtain estimates of the sensitivity of relative growth to these three climate variables (i.e., regression slopes, β's), which are well-established as variables influencing tree growth in the southwestern United States[5,10,13]. Each regression was associated with a coefficient of determination ($R^2$), another metric of the strength of the relationship between climate and tree growth.

We compared these statistics—standard deviation, regression slopes, and coefficient of determination—by contrasting the median value of each across all ITRDB time series at a given sampling site against the median value across all FIA time series within a radius of 100 km of the focal ITRDB site (ITRDB minus FIA). Thus, the comparison is geographically local. Significance of these contrasts was tested using a bootstrapped two-sided Wilcoxon test (1000 iterations) to control for differences in the number of time series between the ITRDB vs. FIA collections. We discarded those ITRDB locations with less than ten FIA samples of the same species within a radius of 100 km. Using climatic water balance (precipitation minus potential evapotranspiration) instead of precipitation and water vapor pressure deficit instead of mean maximum temperature on a subset of the data did not alter the conclusions presented herein.

We conducted an additional analysis that evaluated possible drivers of growth variability per species across the full dataset (i.e. not a local comparison). We used a linear mixed-effects regression approach that modeled variation in the SD of detrended ring-width time series as a function of age (mean cumulative ring count during the 1930–1995 period), database (i.e. ITRDB vs. FIA), and geographic as well as climatic parameters:

$$SD \sim database + \ln(age) + latitude + longitude + elevation +$$
$$mean\ cool\ season\ precipitation + mean\ dry\ season\ mean\ maximum\ temperature.$$
(3)

The mean climate parameters cool season precipitation (previous November to current year March), and dry season mean maximum temperatures (mean of previous year (August to October), and summer of the current year (May to July)) were calculated over the 1930–1995 period. The model included all possible two-way interactions and accounted for repeated values per site via site random effects (modifying the intercept). Model comparison and selection was performed using the dredge function in the R package MuMIn[44]. We selected the best model based on the Akaike Information Criterion (AIC). If multiple models were within two units of AIC, we chose the model with the lowest number of parameters. This analysis was repeated using a negative exponential, a 30-year spline, and a 100-year spline to detrend the time series—each with and without prewhitening—to test whether differences in detrending method or temporal autocorrelation influenced the results of the analysis. We also analyzed variation in the three partial regression slopes (β's) and explained variance ($R^2$) of the regressions predicting relative tree growth as a function of climate, using the same model shown in Eq. 1 (Supplementary Figure 3). We report results from this analysis only based on time series detrended with a negative exponential curve, as there was no notable effect of the choice of detrending on predicting SD. All statistical analysis was carried out in R using the packages dplR[45] and lme4 [46].

**U.S. Southwest aggregation**. For the projection of future growth across the U.S. Southwest, we included ITRDB and FIA tree-ring time series south of 38°N, mirroring the study area of Williams et al.[5], after discarding the ITRDB locations with less than 10 FIA samples of the same species within a radius of 100 km. We then averaged all ITRDB time series into a regional chronology by applying a geographically weighted mean, which takes into account the mean distance of each site to all other sites, giving more weight to samples from areas with few chronologies. Geographic weighting was calculated and applied in each year to account for varying site replication over time, and the resulting regional chronology was variance-stabilized[47]. FIA time series, 2923 of which (out of a total of 2926) fall within at least one 100 km radius around an ITRDB site, were treated following the same procedure. The variance of regional-scale average climate time series was not adjusted. To ensure sufficient sample replication in the most recent years, the analysis included the 1902–2008 period. Specifically, the regional aggregations were based on 208 ITRDB sites (8305 samples) with 40 sites (1282 samples) present in the year 2008, and 1247 FIA sites (2926 samples) with 86 sites (1025 samples) present in 2008.

We then used the following multiple linear regression to predict tree growth variation in the two regional chronologies as a function of precipitation, temperature, and database (ITRDB vs. FIA):

$$ring\ width\ index_t = \ln(precip)_t \times database + tmax_t \times database,$$
(4)

where $t$ is the year, tmax is the average of the two dry season temperature time series, and "database" is either the factor FIA or ITRDB. Randomly subsampling the time series data underlying the ITRDB regional chronology to equal the FIA sample size of 2926 time series (100 times) resulted in explained variance and regression slopes that were essentially unchanged (SD of 0.005 ($R^2$) and 0.002 (regression slopes) among the 100 replicates).

## Data availability

The International Tree-Ring Data Bank is located at: "ITRDB (https://www.ncdc.noaa.gov/paleo-search/)" (11/03/2017). All other data can be made available upon request. Contact R.J.D. for the FIA, C.D.O. for the Pinaleño Mountains, and C.H.G. for the Navajo Nation tree-ring data. A reporting summary for this article is available as a Supplementary Information file.

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

## Acknowledgements

The authors would like to thank the many people who contributed to and maintained the ITRDB, as well as the field and laboratory crew involved in the establishment of the Interior West FIA tree-ring data network, especially Alex Arizpe, Jacob Aragon, Andrew Gray, and Joey Pettit. We thank the Navajo Forestry Department for permission to include the data from Navajo forest lands in the analysis. We are grateful to David Frank for critical discussions of this work. Our thanks also go to Nichole Casebeer and Cara Gibson for their help in creating Fig. 1, and to Ben Hickson for assistance with Figs. 2 and 4. S.K. acknowledges the support of the USDA-AFRI grant 2016-67003-24944; M.E.K.E. was supported by the University of Arizona College of Science and Institute of the Environment. C.H.G. is supported by an EPA STAR Fellowship (#F13F51318) and a grant from the Navajo Nation (contract CO1142).

## Author contributions

S.K., R.J.D., and M.E.K.E. conceived the study and led the writing of the paper with critical inputs from J.D.S., C.H.G., C.D.O., and A.M.L. C.H.G., C.D.O., and A.M.L. contributed additional (unpublished) tree-ring data.

## Additional information

**Competing interests:** The authors declare no conflict of interest.

