## [Peer Review File · Nature Communications]

Reviewers' comments:

Reviewer #1 (Remarks to the Author):

The work assembles an impressive amount of tree-ring data from forest inventory programs (FIA) in the western interior U.S.. Using these data, the authors evaluate whether or not past findings about projected future forest growth undertaken in analyses using the International Tree-Ring Databank (ITRDB) (e.g. William et al. 2013; Charney et al. 2016) were biased by the sampling design inherent to the ITRDB. The main idea is that previous work would have relied on calibration of models using ITRDB data that are not representative of the full ecological spectrum, thus leading to biased projections. Although I see some concern in this work, I greatly appreciated reading the manuscript. My review below follows the logistics of what I see as two separate questions. The first is whether the authors present convincing evidence of biases in the projections of future growth trajectories presented in previous work. Then, is the conclusion of these earlier studies affected by the sampling biases. Intuitively, my answers to these two questions is 'no' for the reasons that follow.

The first observation authors made is that the FIA tree growth data were less sensitive to climate than the ITRDB. But the lower sensitivity in FIA tree growth could arise from a higher degree of autocorrelation in these data, as opposed to the ITRDB (which is somewhat what we can interpret from Fig. S4, right(?)). I do not think that authors have made corrections to the data (whitening) so to ensure that FIA and ITRDB were sharing similar levels of autocorrelation when performing the analysis of climate sensitivity. If the autocorrelation is higher in the FIA data, then that may well explain the lower coefficients and R-squares with the climate variables obtained with these data (perhaps because in very productive forests trees may benefit from greater pools of non-structural carbohydrates and the turnover rates of needles and fine roots may be slower (?)). The less important growth decline noted with the FIA could just be an artefact of the poorer predictive skills of the FIA regression model. I recommend carrying an additional analysis by applying whitening procedures to both datasets to ensure that autocorrelation is not the source of discrepancy (which would ultimately affect the conclusion of this work).

Note also that the model used to project future growth is quite overly simplistic. By computing an average chronology for calibration, the variance of individual trees and sites is eliminated.

The next point is whether the conclusions of earlier studies are affected. Obviously, as stated by the authors in their discussion they are not and the projections of this work strongly concur with the conclusions of William et al., and Charney et al. derived from the ITRDB. The findings from this work are largely supportive of William et al. and others. Hence, the conclusions of earlier work were not dependent on the sampling. In this regard, I find the title and abstract overly stated.

There is perhaps also an issue with the demography of the sampled trees, with those of the ITRDB being old trees as opposed to those sampled in the FIA. This would probably be interesting to see on Fig 1 (additions of bars for range in age and size (?)), which is important considering that old/large trees are often more dependant on water availability or have more constraints.

Aside from that, the paper is well written. But I think it would make a better paper if it was not aimed at challenging past studies but rather as a very constructive evaluation of climate change impacts in the most productive forests examined through the analysis of new exhaustive dataset that is representative of forest stands. This would require only minor editorial revisions. As presented, my feeling is that the paper is aiming at dendrochronologists and not to the wider field.

More specific comments

Line 173-174 Reading this sentence, it gives me the impression that the tree growth series of the ITRDB are more smoothed than those of the FIA. Clearly, in a regression the presence of autocorrelation will affect the regression slopes and the amount of explained variance, and at the end the outcomes (lines 202-204). Please verify that autocorrelation is not affecting the outcomes (by perhaps whitening the data prior to the regression analysis (?)).

Lines 207-225 To my opinion some parts of the discussion are "rough". There is certainly a way of having the same message, but in a more "politically correct manner"?

Figure. I do not know if it is the gray-scale scheme but I did not find Figure 1 highly supportive of

the statement made on lines 60-61 in regard to the distribution of environmental conditions (it seems right for PSME, but not convincing for others). As for ring-width variability, this could be influenced by tree age. Because the ITRDB is essentially a collection of very old trees as opposed to the forest inventories, the ring widths will be smaller.

Figure 5: The number of samples is not clearly indicated by the shaded area curves. Is the number of samples from ITRDB approximating 8000 samples? It would be better if confidence intervals were presented instead of boxes and whiskers, so to see if the projections are significantly different or not.

The Supplementary info is missing a figure showing inter-tree correlation (r_{bar}) at the plot or some relevant scale. Notably, I would like to see if the r_{bar} of the ITRDB chronologies is similar to that of the sampled FIA series across the 100km radius. This could be carried on detrended and whitened data over the period used for analysis of climate sensitivity.

Reviewer #2 (Remarks to the Author):

Review: Klesse et al. . "Overestimation of forest vulnerability to climate change revealed by forest inventory data" Nature Communications

This paper evaluates the difference in growth responses in trees selected for dendroclimatic sensitivity and trees sampled more randomly for forest inventory purposes. It finds, not surprisingly, that trees sampled for dendroclimatic purposes are more sensitive to climate than other trees, and that therefore estimates of vulnerability from the more sensitive trees should not be applied to the full population. The paper's chief contribution is a quantification of these differences for three different species with wide climatic and geographical amplitude in the western US and how the sensitivities vary in space.

The paper is a good contribution to the literature, but it fails to address one significant caveat that needs to be addressed because it is a plausible alternative explanation. Specifically, even for a given detrending method, age-dependent growth and age-related sensitivity to given climatic anomalies have been observed in tree rings (e.g., Szeicz and MacDonald 1994 and papers that cite it, of which many – at least 10, report age-dependent growth-climate relationships for a range of biomes, some water limited, others not). I strongly suspect that the ITRDB data presented in this paper tends to be from older (perhaps much older) trees than that from the FIA, but I could not find this information in the paper, so cannot evaluate it. Without an analysis of this, or a coherent argument why it does not matter despite the literature on age-dependent growth, I do not think the paper is publishable on its own. The authors, in any case, need to refute the age-dependence as a potential determinant of their results, rather than assuming it is due to sampling – the argument central to the paper, e.g. that vulnerability has been systematically overestimated by using the ITRDB, depends on it.

Minor comments:

Line 34: "outrange" → exceed

Line 35: How many past centuries? The statement as written suggests it's infinite, which isn't true. Tree ring data and other proxies give us at best a few millennia of annual resolution.

Line 37-39. The statement that increasing temperature and increasing evaporative demand will increase the frequency, duration, and intensity of droughts is not universally true, and not quite universally true for semi-arid biomes as implied – Cook et al show that it is mostly true for "arid" regions, which in their maps includes quite a bit of semi-arid landscape. It might pay to be more precise here. Also, is reference 6 really the most appropriate here?

Line 39: Reference 9 shows a decline in NPP, not tree growth. Tree growth might be inferred, but NPP includes a lot more than tree growth.

Line 39-41: Decreasing tree growth is associated with, but not a cause of, increasing vulnerability to insect attack. Trees that grow less in the same stand may be more vulnerable, but trees that grow less in general aren't necessarily more vulnerable to insects. Suggest reframing this to be more precise.

Line 41-43: Suggest citing Abatzoglou and Williams PNAS paper here rather than Westerling et al.; more attribution and more adept treatment of water vs. energy balance. Westerling is a detection paper, and does not separate water and energy balance well, focusing more on warming. Also, "Drier conditionsare", not "Drier conditions....is"

Line 195: Depending on who you ask, CMIP5 model estimates are not "predictions" because they have no error, nor are they forecasts, which require a probability estimate. They are more appropriately "projections", if-then estimates given knowledge of the system and its forcings. Therefore predictions of impacts should also be characterized as projections.

Reviewer #3 (Remarks to the Author):

Klesse et al present much-needed evidence that tree-ring data obtained from studies that aim to reconstruct past climate variability (or otherwise maximize climate signals in tree rings; 'targeted sampling') overestimate the climate sensitivity of the same species in the same region that is obtained using a sampling design targeted at representing species' habitat range ('representative sampling'). The study delivers an important contribution to climate change studies and forest ecology, as it finally quantifies the overestimation of climate sensitivity based on targeted sampling. This is highly relevant as most studies estimating this sensitivity use data from targeted sampling, without considering or correcting for this potential overestimation. For a long time – as the authors rightfully mention – we know that overestimation is present and perhaps significant, but so far no study has quantified this. In addition, the study also produces important insights into the environmental conditions leading to larger overestimation.

The analyses are based on a massive amount of data: published tree-ring data of studies using 'targeted' sampling (from the ITRDB database) and – quite unique – tree-ring data from 'representative sampling' of forest inventories (from FIA database), for three common species in the Western US. Data analysis mostly follows standard practices in dendrochronology. The ms is well written and easy to follow. Figures are clear but also quite information-dense, so perhaps need some simplification to better convey the key messages.

I have three main comments:

- There are two important questions with regard to the overestimation problem: (1) Do 'targeted sampling' studies significantly overestimate climate sensitivity of tree growth, and by what percentage? (2) If so, what is then our best estimate of climate sensitivity, without the bias of 'targeted sampling'? In their study, the authors tackle question 1 by comparing variance and climate sensitivity of 'targeted' and 'representative' sampling, making sure that the ITRDB and FIA tree-ring data are obtained from the same area. (That is, within a 100-km radius). And they find evidence for significant and large overestimation of climate sensitivity. The selection of sites to tackle Question 1 is crucial: this corresponded to the sites in the ITRDB database. This implies that

an analysis of climate sensitivity based on the FIA database will still not be representative for the study species. It is limited to the ITRDB sites and it is very unlikely (due to the aim of targeted sampling studies, logistical constraints, etc) that these sites cover the distribution area and do so in a representative way (this is clearly illustrated in Fig 1; I assume that all FIA sites –so including the ones not used in the study – are shown here). As a result, it seems that – the analyses of the climate sensitivity and climate-change projections for the ‘representative sampling’ is still not completely representative for the study species. The solution would be to include all or a really representative selection of FIA tree-ring data to perform an analysis of climate sensitivity, very much in the same way as the authors now did on the dataset constrained by ITRDB sample sites.

- Climate sensitivities in ITRDB and FIA datasets are compared in an indirect way, by building local or regional chronologies for both datasets, and then comparing R², partial regression coefficients and slopes (using Wilcoxon paired tests). Yet, this comparison is indirect and not ideal to test the effect of source database (ITRDB or FIA) on climate sensitivity. It would be better to directly test the modulating effect of source database on climate sensitivity by combining both data into one analysis (per ITRDB site, or regionally) and then testing for significant interactions between source database and climate variable, for all climate variables included in the tests. This is the appropriate statistical test to tell whether climate sensitivity differs between the two sources of data, as it combines the data, and directly tests the effect of ‘database’. For the effect of database on R², one could consider to calculate R² change when starting with FIA data and adding ITRDB data; or vice versa. Less straightforward, but also here it’s better to test this in a direct way, within a statistical test, instead of a statistical test of the database effect that uses the output of two separate tests.
- In their final set of analyses, the authors use the climate sensitivity obtained with regional chronologies for ITRDB and FIA data to project ring-width index until the end of this century, ‘forcing’ (or feeding) their regression models with CMIP5 climate change projections. Such projections are tricky and need to be interpreted with much care. (The authors do warn for “over-interpreting” the zero growth for the ITRDB chronology) First, climate effects are extrapolated beyond the range of temperatures during the period for which the chronology was built. It is unlikely that coefficients in the regression models will remain unchanged at higher temperatures. Second, I am hesitant to interpret sensitivity of tree-ring width to fluctuations in temperature and rainfall, as the response of tree growth to a gradual increase in temperature over time. Finally, I also think this extrapolation is not needed: the quantification of the overestimation of sensitivity already make this an important and much-needed study.

Other comments:

- I wonder whether the ITRDB chronologies a subset of all published chronologies, and if so, whether (and how) that subset is then biased... Doesn’t thi s
- A radius of 100 km was applied around the ITRDB site to obtain tree-ring data from FIA. How robust are results to changes in this radius? I understand that this radius is a compromise between sample size and representability, and decreasing the radius would decrease sample size and statistical power, but given the mountainous habitat of these tree species, a radius of 100 km may include a huge variation of environmental conditions.
- What was done with overlapping 100-km radii? So, in case ITRDB chronologies were <200 km apart. This could have led to a re-use of FIA ring-width series, which is clearly not desirable.
- Structure of the Supplementary material. This structure is not so clear and not very logical. Explanation of statistical tests (1.3) precede text on data availability (1.4) and sampling design (3).
- It was unclear to me why the climate sensitivity analysis was only performed for the southwest. This is also not explained in the section 1.4 of the Appendix. (In that section, reference is made to Williams ref 16, but this appears as 15 in the main text.)
- I missed advices for users or contributors or the ‘management’ of the ITRDB on how to avoid, correct or deal with this overestimation.
- I also missed some general notes on whether this overestimation is to be expected also in other forest types, regions, species.
- The authors call for more tree-ring data samples in a representative way: this is evident. But could they also provide more concrete advice on tree-ring sampling during forest inventories?
- This study considers site selection as the main way in which ‘targeted sampling’ is conducted.

But this is just one of the ways at which dendrochronologists/climatologists attempt to increase the climate (or common) signal in tree-ring data. Other steps include: selection of trees to be sampled (dominant), selection of tree-ring series to be included in analyses; selection of series in master chronology; selection of chronologies to be uploaded to the ITRDB (!). What were the effects of the other steps? And was the selection of dominant trees in the ITRDB studies done in a similar way as in the FIA studies?

We thank the three reviewers for their detailed and constructive comments. The
reviewers commented positively on the importance of comparing climate sensitivity of
“targeted” vs. forest inventory-based tree-ring samples – i.e., the ITRDB (International
Tree-Ring Data Bank) vs. FIA (Forest Inventory and Analysis) comparison – with respect
to its implications for projected future tree growth under future climates. The primary
concerns were (very briefly summarized): Reviewer 1 asked about differences in
temporal autocorrelation between the two sets of time series, and secondarily raised the
issue of differences in the age of the sampled trees; Reviewer 2’s concern was age
differences between the ITRDB vs. FIA samples; Reviewer 3 wanted to see a
comparison made between ITRDB vs. all FIA samples (not just those FIA samples
located within a 100-km radius of an ITRDB sampling site), and wanted to see a
database effect (ITRDB vs. FIA) tested in a single model with all the data.

Beginning with Reviewer 3, we addressed her/his comments with a new analysis – a
linear mixed effects model that attributes differences in growth variability to the effects of
age, mean annual temperature and precipitation, elevation, latitude, longitude, and
database (i.e. FIA vs. ITRDB). The inclusion of age as a factor in this linear mixed
effects model addresses the primary point raised by Reviewer 2 (and secondary point
raised by Reviewer 1). Finally, to address Reviewer 1’s point about autocorrelation, we
ran this model with and without prewhitening (i.e. the removal of autocorrelation in the
individual time series) and 3 different methods of detrending.

The reviewer’s comments converged to a large degree (particularly regarding age
effects on climate sensitivity) and the analysis they stimulated us to conduct added a
new dimension to the manuscript: we now not only quantify a) the difference of growth
variability and climate sensitivity between the ITRDB and FIA collections and b) the
consequences of contrasting climate sensitivity (with respect to future tree growth), but
we also c) evaluate the reasons why climate sensitivities differ between the two
collections. The three hypotheses (candidate explanations) that we evaluate are age,
macrosite selection (i.e., mean climate, captured in terms of climate normals, elevation,
latitude, longitude), and microsite selection (i.e., a database effect after correcting for all
other factors). We find evidence in support of these three explanations that varies
across species, and environmental and geographic gradients. Overall, the reviewers’
comments led to a much improved manuscript, exploring the reasons *why* the two
collections differ in climate sensitivity, adding depth to the overarching message of our
manuscript, that forest growth in the southwestern United States will decline, but to a
significantly lower degree than previously reported.

We also offer a new title for the manuscript, to more clearly focus the scope of inference
on the U.S. Southwest, which is the domain of most of the analyses, and in particular
the projection of future tree growth. This change was made in response to a comment
from Reviewer 1. We are flexible though, and leave it to the reviewers and editor to
decide between the old vs. new titles for the manuscript.

Please find below point-by-point responses to the individual comments of the reviewers.

Reviewer #1 (Remarks to the Author):

The work assembles an impressive amount of tree-ring data from forest inventory
programs (FIA) in the western interior U.S.. Using these data, the authors evaluate
whether or not past findings about projected future forest growth undertaken in analyses
using the International Tree-Ring Databank (ITRDB) (e.g. William et al. 2013; Charney
et al. 2016) were biased by the sampling design inherent to the ITRDB. The main idea is
that pervious work would have relied on calibration of models using ITRDB data that are
not representative of the full ecological spectrum, thus leading to biased projections.
Although I see some concern in this work, I greatly appreciated reading the manuscript.

Thank you!

My review below follows the logistics of what I see as two separate questions. The first
is whether the authors present convincing evidence of biases in the projections of future
growth trajectories presented in previous work. Then, is the conclusion of these earlier
studies affected by the sampling biases. Intuitively, my answers to these two questions
is 'no' for the reasons that follow.

The first observation authors made is that the FIA tree growth data were less sensitive
to climate than the ITRDB. But the lower sensitivity in FIA tree growth could arise from a
higher degree of autocorrelation in these data, as opposed the ITRDB (which is
somewhat what we can interpret from Fig. S4, right(?)). I do not think that authors have
made corrections to the data (whitening) so to ensure that FIA ad ITRDB were sharing
similar levels of autocorrelation when performing the analysis of climate sensitivity. If the
autocorrelation is higher in the FIA data, then that may well explain the lower coefficients
and R-squares with the climate variables obtained with these data (perhaps because in
very productive forests trees may benefit from greater pools of non-structural
carbohydrates and the turnover rates of needles and fine roots may be slower (?)).
I recommend carrying an additional analysis by applying whitening procedures to both
datasets to ensure that autocorrelation is not the source of discrepancy (which would
ultimately affect the conclusion of this work).

We thank the reviewer for this point regarding the potential influence of auto-correlation
on our results. In response, we have run additional analyses with the individual tree
time-series pre-whitened. In particular, we reevaluated climate sensitivities (regression
slopes with respect to climate predictors, shown in Figure 4 in the manuscript) with vs.
without pre-whitening. We include below Figure 4 with vs. without pre-whitening.
While there are some subtle differences, the results remain essentially unchanged. We
have added this information to the text and methods.

Figure 4 with all series pre-whitened prior to the analysis:

Figure 4 as is in the manuscript:

In addition, we conducted a new analysis (a linear mixed-effects model) of the possible factors explaining variation in the standard deviation (SD) of detrended ring width time series, including with vs. without pre-whitening. We do not see differences between the results of the linear mixed effects models when the time series data are vs. are not prewhitened. Please see a more detailed description below (175-226), since this new analysis was conducted in response to Reviewer 1’s comment about the effect of age (or size) on climate sensitivity.

The less important growth decline noted with the FIA could just be an artefact of the poorer predictive skills of the FIA regression model.

Please note that the difference in explained variance between the regional-scale FIA vs. ITRDB regressions is only 4% – 64.7% variance explained in the case of the ITRDB

sample vs. 60.7% variance explained in the case of the FIA sample. This, combined with
the strong difference in the values of the regression coefficients (slopes 59% and 41%
steeper with respect to climate predictors based on the ITRDB sample), leads us to the
conclusion that the reduced growth decline of FIA trees in Figure 5 is not an artifact of
the difference in skill between the two regressions.

Note also that the model used to project future growth is quite overly simplistic. By
computing an average chronology for calibration, the variance of individual trees and
sites is eliminated.

We agree that the model to project future tree growth is simple. First, however, keep in
mind that this model was chosen in order to replicate the methods of Williams et al.
(2013), so that we could compare their results to what we get based on a new,
representative sample. Second, the model chosen by Williams et al. (2013, i.e. the
climatic variables used as predictors) is well justified – there is no mystery regarding
what the important climate drivers of tree growth are in the southwestern U. S.; this is
well documented in a long series of publications (e.g., Fritts et al. 1965. Fritts 1976,
Adams & Kolb 2005, Williams et al. 2010, to name a few).

We disagree with the second point and want to remind the reviewer that by computing
an average chronology, the chronology's variance is a result of the variance of the
individual samples, the between-sample correlation, and the number of series. Since we
performed variance stabilization that accounts for differences in sample replication and
between-sample correlation, the main factor left to influence the final chronology
variance is the original variance of individual samples contributing to each chronology.

The next point is whether the conclusions of earlier studies are affected. Obviously, as
stated by the authors in their discussion they are not and the projections of this work
strongly concur with the conclusions of William et al., and Charney et al. derived from
the ITRDB. The findings from this work are largely supportive of William et al and others.
Hence, the conclusions of earlier work were not dependent on the sampling. In this
regard, I find the title and abstract overly stated.

We note that there are two ways in which our results might differ from those of Williams
et al. (2013) and Charney et al. (2016): in sign and in magnitude. We agree with the
reviewer that our results do not differ from the previous studies in sign – tree growth in
the Southwest will decline. But there is a striking difference in terms of magnitude: it is
clear in Figure 5 that there is a strong difference in future projected tree growth by the
2070-2090 time frame if FIA vs. ITRDB samples are used for calibration and prediction.
A 42% greater reduction in future growth (by 2070-2099) is not small; the conclusions of
previous projections of future growth *do* depend on the sample, and in particular, the
climate sensitivity of the sample that was used. The forest inventory sample better
characterizes the distribution of climate sensitivity, because it is a sample that is
specifically designed to be representative.

There is perhaps also an issue with the demography of the sampled trees, with those of
the ITRDB being old trees as opposed to those sampled in the FIA. This would probably
be interesting to see on Fig 1 (additions of bars for range in age and size (?)), which is

important considering that old/large trees are often more dependant on water availability
or have more constraints.

This is an excellent point, which was also raised by reviewer #2. In response, we ran a
new analysis that evaluates simultaneously many possible causes of variation in growth
variability, including age, latitude, longitude, elevation, climate normals (cool season
precipitation and dry season maximum temperature), as well as database (FIA vs.
ITRDB). That is, we ran a multiple regression (linear mixed-effects) model that examines
variation (among tree-level time series) in the SD of detrended ring-widths as a function
of tree age (defined as the mean cumulative ring count during the 1930-1995 period),
database (i.e. ITRDB vs FIA), and geographic as well as climatic parameters, as follows:

$SD \sim Database + \log(\text{age}) + \text{lat} + \text{lon} + \text{elevation} + \text{meanWinterprecip} +$
$\text{meanTmaxdryseason},$

The model included all possible two-way interactions and accounted for repeated
samples (i.e., trees) per site (via site random effects modifying the intercept). We ran
that model for time series detrended using a negative exponential function, a 30-year
spline, and a 100-year spline – both with and without whitening – with very similar
results across these six methodological variants (see Fig S1).

In all three species, age has a significant, positive effect on the SD of detrended ring
widths, confirming the reviewer's point – the growth of older (larger) trees is more
climate sensitive. In the case of two species, *Pinus ponderosa* and *Pinus edulis*, there
remains a significant, negative database effect - meaning FIA time series have
significantly lower SD, after accounting for age and geographic and climatic parameters.
The one species (Douglas-fir, *Pseudotsuga menziesii*) that does not show a significant
database effect has a significant interaction effect between database and mean cold
season precipitation, i.e. Douglas-fir trees sampled in forest inventory plots have
significantly lower SD (than ITRDB samples) in wetter regions, implying that in relatively
mesic places, ITRDB Douglas-fir samples were likely chosen from special, drier-than-
average microsites (e.g., steeper, less available soil water capacity), as illustrated in
Figure 1b. Further, we show in Figure S1 that there are significant interaction effects
between database and latitude, and database and longitude or elevation, for *Pinus*
*ponderosa* and *Pinus edulis*. This summary graph showing the effect sizes from the
multiple regression, for all three species and six different combinations of
detrending*prewhitening, has been added to the supplementary material (Fig. S1).

For Douglas-fir (*Pseudotsuga menziesii*), we conclude the difference in climate
sensitivity between FIA and ITRDB samples is caused by (a) older trees on average in
the ITRDB sample than in inventory plots, (b) sampling at lower (thus warmer)
elevations, that are not representative of the elevational distribution of Douglas-fir, and
(c) a microsite sampling bias that is evident at more mesic locations (significant
interaction effect between database and mean cold season precipitation).
We do not see evidence of targeted sampling of *Pinus ponderosa* and *Pinus edulis* trees
at lower elevation in the ITRDB sample compared to their elevation distribution in the
FIA sample (see Fig. 1a). Instead, the multiple regression analysis indicates that the
difference in sensitivity (SD of detrended ring widths) is caused by a significant age

effect (ITRDB trees are older) and likely the effect of targeting trees at microsites with
high climatic sensitivity, as illustrated in Figure 1b (i.e., the significant effect of database,
after controlling for age, elevation, other main effects and 2-way interactions).

Note also that we have exchanged the density strips for standard deviation in Figure 1b
and show now the distribution of age of the three species and sample collections, similar
to the elevation density strips of Figure 1a.

Aside from that, the paper is well written. But I think it would make a better paper if it
was not aimed at challenging past studies but rather as a very constructive evaluation of
climate change impacts in the most productive forests examined through the analysis of
new exhaustive dataset that is representative of forest stands. This would require only
minor editorial revisions. As presented, my feeling is that the paper is aiming at
dendrochronologists and not to the wider field.

Thank you. Our intent was not to challenge past studies, rather to update them in the
light of newly available data. The issue we raise is not one those authors were unaware
of; Williams et al. (2010, 2013) and Charney et al. (2016; including the same senior
author as the current manuscript, M. E. K. Evans) explicitly advised caution concerning
the magnitude of predicted trends due to the potential bias of targeting climate-sensitive
trees. Our goal here is to advance the field by providing a quantitative estimate of the
bias that others have suspected was there (due to standard sampling practices of
dendrochronology – the “site and tree selection” principles), and report the magnitude of
overestimation of growth decline.

We aim not at dendrochronologists, but at the wide audience concerned about forest
health and the carbon sink strength of terrestrial ecosystems (of which forests are a very
important component). As evidence of this point, we note that the Williams et al. (2013)
paper has been cited 333 times as of this writing (and is considered a “highly-cited
paper” by ISI), making it all the more important to put their findings (as well as Williams
et al. 2010, with 261 citations, and Charney et al. 2016, with 28 citations) in context.
That is, we view our contribution as a constructive evaluation of climate change impacts
based on a dataset that is representative of the full ecological spectrum of PSME, PIPO,
and PIED forest stands. Having said that, we have revisited the discussion to reframe
and rephrase our conclusions in as positive and diplomatic language as we can – we
emphasize that dendrochronologists in the past had different research questions and
their sampling practices were successful with respect to maximally addressing those
research questions. We welcome any specific suggestions if there remain passages that
strike the reviewer as “rough” (as per the comment below).

More specific comments

Line 173-174 Reading this sentence, it gives me the impression that the tree growth
series of the ITRDB are more smoothed than those of the FIA. Clearly, in a regression
the presence of autocorrelation will affect the regression slopes and the amount of
explained variance, and at the end the outcomes (lines 202-204). Please verify that
autocorrelation is not affecting the outcomes (by perhaps whitening the data prior to the
regression analysis (?)).

The auto-correlative properties of the two regional chronologies are very similar.

The significant partial autocorrelation coefficients for the two regional chronologies are
lag 1 and 2:

	AR1	AR2
ITRDB	0.368	0.270
FIA	0.408	0.241

We have added these numbers to the manuscript in lines 203-204.

Please see the response above (lines 81 and following, and 175-226), which details the
new analyses demonstrating that removing autocorrelation does not change the results
and conclusions.

Lines 207-225 To my opinion some parts of the discussion are “rough”. There is
certainly a way of having the same message, but in a more “politically correct manner”?

Please see the detailed response to a similar comment above, lines 234-257.

Figure. I do not know if it is the gray-scale scheme but I did not find Figure 1 highly
supportive of the statement made on lines 60-61 in regard to the distribution of
environmental conditions (it seems right for PSME, but not convincing for others). As for
ring-width variability, this could be influenced by tree age. Because the ITRDB is
essentially a collection of very old trees as opposed to the forest inventories, the ring
widths will be smaller.

The caption of Figure 1 indicates that elevation bias affects only PSME. In the revision,
we have added a paragraph in the discussion (lines 262-280) highlighting the effects of
macro-site selection (elevation), age, and micro-site conditions on growth variability for
all three species and what it means in terms of sample bias of the ITRDB in the US
Southwest. We have also exchanged the density strips for growth variability in panel b)
with density strips for age.

Figure 5: The number of samples is not clearly indicated by the shaded area curves. Is
the number of samples from ITRDB approximating 8000 samples? It would be better if
confidence intervals were presented instead of boxes and whiskers, so to see if the
projections are significantly different or not.

Regarding the first point, yes, there are ~8000 samples available (and used in the
analysis) from the ITRDB for our three study species.

Regarding the second point, the variability in future tree growth represented by boxplots
derives from variability among 15 GCMs, i.e., in projected mean climate during the
2010-2039, 2040-2069, and 2070-2099 periods. For projection of future tree growth, we
have used one value (of projected future mean climate) per GCM per time window. The
boxplots are intended to illustrate the possible range of growth declines deriving from
uncertainty about future climate conditions (driver uncertainty). We have added asterisks

to Figure 5 to show whether the distribution of 15 values per time period differs
significantly when calibration and projection is based on the ITRDB vs. FIA samples.

The Supplementary info is missing a figure showing inter-tree correlation (r_{bar}) at the
plot or some relevant scale. Notably, I would like to see if the r_{bar} of the ITRDB
chronologies is similar to that of the sampled FIA series across the 100km radius. This
could be carried on detrended and whitened data over the period used for analysis of
climate sensitivity.

The reported standard deviation and climate sensitivities are for the median tree
response at each ITRDB site and the median response of the FIA samples within a 100
326 km radius around the ITRDB site. SD and climate sensitivity are not calculated on the
327 chronology level, where r_{bar} would be indeed a useful metric to show mean inter-series
correlation. We have added now information about r_{bar} for the southwestern US sample
pool (south of 38 degrees N) in the text (ms line 216): the inter-series correlation is
greater for the ITRDB sample (0.328) compared to the FIA sample (0.202). This too is
consistent with the fact that traditional dendrochronology and dendroclimatology
sampling was aimed at maximizing climate signal common across trees and minimizing
other sources of variation in ring widths that are idiosyncratic to an individual tree. Yet,
the variance in (detrended) ring widths explained by climate predictors is almost the
same between the two regional chronologies – despite the differences in r_{bar} .

Reviewer #2 (Remarks to the Author):

Review: Klesse et al. . “Overestimation of forest vulnerability to climate change revealed
by forest inventory data” Nature Communications

This paper evaluates the difference in growth responses in trees selected for
dendroclimatic sensitivity and trees sampled more randomly for forest inventory
purposes. It finds, not surprisingly, that trees sampled for dendroclimatic purposes are
more sensitive to climate than other trees, and that therefore estimates of vulnerability
from the more sensitive trees should not be applied to the full population. The paper’s
chief contribution is a quantification of these differences for three different species with
wide climatic and geographical amplitude in the western US and how the sensitivities
vary in space.

The paper is a good contribution to the literature, but it fails to address one significant
caveat that needs to be addressed because it is a plausible alternative explanation.
Specifically, even for a given detrending method, age-dependent growth and age-related
sensitivity to given climatic anomalies have been observed in tree rings (e.g., Szeicz and
MacDonald 1994 and papers that cite it, of which many – at least 10, report age-
dependent growth-climate relationships for a range of biomes, some water limited,
others not). I strongly suspect that the ITRDB data presented in this paper tends to be
from older (perhaps much older) trees than that from the FIA, but I could not find this
information in the paper, so cannot evaluate it. Without an analysis of this, or a coherent
argument why it does not matter despite the literature on age-dependent growth, I do
not think the paper is publishable on its own. The authors, in any case, need to refute

the age-dependence as a potential determinant of their results, rather than assuming it
is due to sampling – the argument central to the paper, e.g. that vulnerability has been
systematically overestimated by using the ITRDB, depends on it.

We thank the reviewer for this excellent point, which converges on the issue of the age
of ITRDB vs. FIA trees, raised by reviewer #1 as well. We added information about the
distribution of age of ITRDB vs. FIA trees in the text and Figure 1 panel b). As detailed
above (lines 175-226), we have added a new analysis that evaluates the effect of age
on growth variability, using a multiple regression (linear mixed effects) model. The age
effect is indeed present, for all three species. We still see an effect of database (i.e. the
contrast ITRDB vs. FIA) on climate sensitivity (the standard deviation of detrended ring
width time series) for *Pinus ponderosa* and *Pinus edulis*, which we hypothesize is due to
the kind of targeted micro-site selection illustrated in Figure 1b. Regardless of the
reasons why climate sensitivities differ between the two samples, they clearly differ, and
this has implications for projected future tree growth under projected future climate
conditions. The fact that ITRDB trees are so clearly much older than FIA trees (Figure
1), and the significant effect of age on climate sensitivity (with older trees showing more
380 year-to-year variability in growth, as suggested by the reviewer), is another reason why
the ITRDB trees are not representative of the actual populations of *Pinus ponderosa*,
*Pinus edulis*, and *Pseudotsuga menziesii* in forests across the interior western U. S.,
and should not be used to make conclusions about future tree growth. Projections based
on ITRDB samples overestimate future forest growth decline in the southwestern United
States, among other reasons, because ITRDB trees are older and more climate
sensitive.

Minor comments:

Line 34: “outrange” → exceed

This has been changed.

Line 35: How many past centuries? The statement as written suggests it’s infinite, which
isn’t true. Tree ring data and other proxies give us at best a few millennia of annual
resolution.

We added the word “several”, meaning roughly the last six hundred to one thousand
401 years.

Line 37-39. The statement that increasing temperature and increasing evaporative
demand will increase the frequency, duration, and intensity of droughts is not universally
true, and not quite universally true for semi-arid biomes as implied – Cook et al show
that it is mostly true for “arid” regions, which in their maps includes quite a bit of semi-
arid landscape. It might pay to be more precise here. Also, is reference 6 really the most
appropriate here?

We clarified that the region of interest in this sentence is the semi-arid areas (as

represented by Fig. 11, Cook et al. 2014)

And yes, Reference 6, as we had it in the references, is clearly a mistake from our side
(wrong Wang et al. 2012 from the literature software used). Thank you for catching this
error. It should be:

Wang, K., Dickinson, R. E. & Liang, S. Global Atmospheric Evaporative Demand over
Land from 1973 to 2008. *J. Climate* **25**, 8353–8361 (2012).

Line 39: Reference 9 shows a decline in NPP, not tree growth. Tree growth might be
inferred, but NPP includes a lot more than tree growth.

This is true. To keep the section streamlined, and to not delve too much into partitioning
components of ecosystem productivity, we changed the reference to Allen et al. 2015
(Ecosphere) and re-use Williams et al. 2013.

Line 39-41: Decreasing tree growth is associated with, but not a cause of, increasing
vulnerability to insect attack. Trees that grow less in the same stand may be more
vulnerable, but trees that grow less in general aren't necessarily more vulnerable to
insects. Suggest reframing this to be more precise.

We have rephrased this sentence to remove the impression of a direct causal
relationship.

Line 41-43: Suggest citing Abatzoglou and Williams PNAS paper here rather than
Westerling et al.; more attribution and more adept treatment of water vs. energy
balance. Westerling is a detection paper, and does not separate water and energy
balance well, focusing more on warming. Also, "Drier conditionsare", not "Drier
conditions....is"

Thank you for the suggestion. We changed the citation.

Line 195: Depending on who you ask, CMIP5 model estimates are not "predictions"
because they have no error, nor are they forecasts, which require a probability estimate.
They are more appropriately "projections", if-then estimates given knowledge of the
system and its forcings. Therefore predictions of impacts should also be characterized
as projections.

This too has been changed, and we appreciate the attention to precise language.

Reviewer #3 (Remarks to the Author):

Klesse et al present much-needed evidence that tree-ring data obtained from studies
that aim to reconstruct past climate variability (or otherwise maximize climate signals in
tree rings; ‘targeted sampling’) overestimate the climate sensitivity of the same species
in the same region that is obtained using a sampling design targeted at representing
species’ habitat range (‘representative sampling’). The study delivers an important
contribution to climate change studies and forest ecology, as it finally quantifies the
overestimation of climate sensitivity based on targeted sampling. This is highly relevant
as most studies estimating this sensitivity use data from targeted sampling, without
considering or correcting for this potential overestimation. For a long time – as the
authors rightfully mention – we know that overestimation is present and perhaps
significant, but so far no study has quantified this. In addition, the study also produces
important insights into the environmental conditions leading to larger overestimation.
The analyses are based on a massive amount of data: published tree-ring data of
studies using ‘targeted’ sampling (from the ITRDB database) and – quite unique – tree-
ring data from ‘representative sampling’ of forest inventories (from FIA database), for
three common species in the Western US. Data analysis mostly follows standard
practices in dendrochronology. The ms is well written and easy to follow. Figures are
clear but also quite information-dense, so perhaps need some simplification to better
convey the key messages.

Thank you.

I have three main comments:

- There are two important questions with regard to the overestimation problem: (1) Do
‘targeted sampling’ studies significantly overestimate climate sensitivity of tree growth,
and by what percentage? (2) If so, what is then our best estimate of climate sensitivity,
without the bias of ‘targeted sampling’? In their study, the authors tackle question 1 by
comparing variance and climate sensitivity of ‘targeted’ and ‘representative’ sampling,
making sure that the ITRDB and FIA tree-ring data are obtained from the same area.
(That is, within a 100-km radius). And they find evidence for significant and large
overestimation of climate sensitivity. The selection of sites to tackle Question 1 is crucial:
this corresponded to the sites in the ITRDB database. This implies that an analysis of
climate sensitivity based on the FIA database will still not be representative for the study
species. It is limited to the ITRDB sites and it is very unlikely (due to the aim of
targeted sampling studies, logistical constraints, etc) that these sites cover the
distribution area and do so in a representative way (this is clearly illustrated in Fig 1; I
assume that all FIA sites –so including the ones not used in the study – are shown
here).

As a result, it seems that – the analyses of the climate sensitivity and climate-change
projections for the ‘representative sampling’ is still not completely representative for the
study species. The solution would be to include all or a really representative selection of
FIA tree-ring data to perform an analysis of climate sensitivity, very much in the same
way as the authors now did on the dataset constrained by ITRDB sample sites.

The distributions of elevation shown in Fig.1a are based on those FIA sites from which
increment cores are available in the states of Arizona, Colorado, New Mexico, and Utah,

i.e. corresponding to the area where the three study species (PIED, PIPO, and PSME)
co-occur. We have clarified this point in the figure caption.

It is true that in creating Figures 3 and 4 – which illustrate comparisons of (Fig 3) the
standard deviation of detrended time series and (Fig 4) regression slopes with respect to
climate – we used only ITRDB sites that have at least 10 FIA samples within a 100km
radius. In these comparisons, 89 % of all FIA Douglas-fir, 95% of all FIA common
pinyon, and 82% of FIA ponderosa pine samples are used. The only significant region
that was excluded in this analysis is eastern Montana; see the distribution of ITRDB
locations in Figure 2.

The second focus of this paper (the formation of a region-wide U. S. Southwestern
growth signal and projection of future tree growth based on the ITRDB vs. FIA samples)
uses virtually all FIA samples of PSME, PIPO, and PIED south of 38°N, as almost all of
them (2923 of 2926) fall within a 100km radius of at least one ITRDB site. In other
words, the FIA sample included in the analysis underlying Figure 5 is virtually complete,
and, we would argue, as close to “unbiased” or “representative” as can be expected,
and certainly much more so than any other existing tree-ring collection in the western U.
S.

The shift in (geographic) focus of the revised title is, in part, in response to this comment
by Reviewer 3. South of 38°N is where we can most comfortably say that ITRDB
samples are being compared against a “representative” sample.

- Climate sensitivities in ITRDB and FIA datasets are compared in an indirect way, by
building local or regional chronologies for both datasets, and then comparing R², partial
regression coefficients and slopes (using Wilcoxon paired tests). Yet, this comparison is
indirect and not ideal to test the effect of source database (ITRDB or FIA) on climate
sensitivity. It would be better to directly test the modulating effect of source database on
climate sensitivity by combining both data into one analysis (per ITRDB site, or
regionally) and then testing for significant interactions between source database and
climate variable, for all climate variables included in the tests. This is the appropriate
statistical test to tell whether climate sensitivity differs between the two sources of data,
as it combines the data, and directly tests the effect of ‘database’. For the effect of
database on R², one could consider to calculate R² change when starting with FIA data
and adding ITRDB data; or vice versa. Less straightforward, but also here it’s better to
test this in a direct way, within a statistical test, instead of a statistical test of the
database effect that uses the output of two separate tests.

This was an excellent suggestion that we took to heart, since putting all the data into a
single statistical test (a multiple regression model) allowed us to simultaneously evaluate
many factors that might influence variability in growth in response to climate, including
age, which both reviewers 1 and 2 pointed out is likely an important factor (and they
were right). It also prevents us from re-using FIA data in the local-scale comparison as
the reviewer pointed out in a comment below. We ran a new analysis of the standard
deviation of detrended (relative) ring-width time series (SD), evaluating age, elevation,

latitude and longitude, climate normals, and finally database identity (FIA vs. ITRDB) as
predictors of variation in relative growth. This analysis included all time series in both the
ITRDB and FIA collections, addressing reviewer 3's concern about "filtering" of some of
the FIA data based on their proximity to ITRDB sampling sites. This indeed provided
valuable new insight into the reasons why climate sensitivities are heightened in the
ITRDB collection, and how this varies across species and environmental and geographic
gradients (climate normal, latitude, longitude, and elevation).

The age effect is present and significant for all three species and we still see a
significant effect of database (i.e. ITRDB vs. FIA) on standard deviation for *Pinus*
*ponderosa* and *Pinus edulis*, which we hypothesize is due to the kind of targeted micro-
site selection illustrated in Figure 1b.

For a more detailed evaluation of the mixed effects model we refer the reviewer to lines
175-226.

We furthermore tested the significance of regression slope differences of the regional
chronologies by a multiple regression including FIA and ITRDB as factors for each of the
two regression slopes in the form of:

Ring width index \sim database*precip + database*tmax.

The conclusion is they are significantly different for both climate predictors. We have
added this description to the methods and the significance level to main text in line 203.

In their final set of analyses, the authors use the climate sensitivity obtained with
regional chronologies for ITRDB and FIA data to project ring-width index until the end of
this century, 'forcing' (or feeding) their regression models with CMIP5 climate change
projections. Such projections are tricky and need to be interpreted with much care. (The
authors do warn for "over-interpreting" the zero growth for the ITRDB chronology) **First**,
climate effects are extrapolated beyond the range of temperatures during the period for
which the chronology was built. It is unlikely that coefficients in the regression models
will remain unchanged at higher temperatures. **Second**, I am hesitant to interpret
sensitivity of tree-ring width to fluctuations in temperature and rainfall, as the response
of tree growth to a gradual increase in temperature over time. **Finally**, I also think this
extrapolation is not needed: the quantification of the overestimation of sensitivity already
make this an important and much-needed study.

1. We agree with the reviewer that climate responses might change as conditions
change – i.e., as the range of temperature or drought stress conditions
experienced by a tree population exceeds the historical range of variability. This
is indeed one of the central challenges of ecological forecasting. The reviewer is
correct that we have advised caution against taking the magnitude of projected
future growth decline literally, for exactly the reasons stated by the reviewer - we
did not account for possible shifts in climate sensitivities, or the effect of CO2
fertilization, etc. The important point is that we are focused on the comparison
between future growth based on the climate-sensitive ITRDB sample vs. the
representative FIA sample, we emphasized this now (in lines 292-295 and 311-

312 of the manuscript). All else being equal, these cautionary notes would apply
equally to both datasets, and thus the comparison remains valid. It may be the
case that in the future, we will understand ways in which these cautionary notes
apply differently to the two samples, which are decidedly different samples; we
would argue that that additional level of refinement belongs to another iteration of
the process of continually improving our projections of future tree growth.

2. Assuming the trees respond to inter-annual variations (after detrending / high-
pass filtering both tree-ring and climate data) in precipitation and temperature, we
would call them sensitive to those parameters (e.g., climate sensitivity is high). If
we allowed more low frequency in both datasets, i.e., don't detrend the climate,
and the ring widths still tracked those two climate variables, gradually decreasing
ring widths with gradually increasing temperatures, (including the year-to-year
fluctuations) why shouldn't we call it sensitivity any more? Furthermore, mean
growth rates in the southwestern United States are lower at low elevations
compared to trees growing at higher elevation. As temperature and elevation are
highly correlated we infer that mean growth rates decrease with increasing mean
temperature. Following the logic to call a regression slope "sensitivity to a unit
change in a predictor" mean growth rates are sensitive to changes in mean
temperatures.

3. We agree with the reviewer that a bias towards climate-sensitive trees has
important implications for ecological forecasting of future tree growth, but not
everyone is able to draw that conclusion from regression slopes that differ
between two samples. We are of the opinion that Figure 5 serves a very
important purpose – visualizing the consequences of heightened sensitivity of the
ITRDB samples, or, put another way, visually communicating the contrast
between future forest growth based upon the climate-sensitive ITRDB vs.
representative FIA collections. The Williams et al (2010 and 2013) papers have
been highly cited –especially because Williams et al. 2013 makes a visual
demonstration of the trend in tree growth implied by a) climate sensitivities
combined with b) projected climate change. Our goal here is to point out that
Williams et al.'s (2010, 2013) very impactful figures need to be adjusted upwards,
because of the known issue of biased sampling of climate-sensitive trees. The
reviewer is of course correct in so many ways that when we make such a figure,
we are at risk of believing what is in the figure. It is in this sense that the
projection of future growth may be viewed as a thought exercise, and it is in this
spirit that we emphasize caution against taking the magnitude of the growth
decline literally (see lines 310-312 in the manuscript).

Other comments:

- - I wonder whether the ITRDB chronologies a subset of all published chronologies,
and if so, whether (and how) that subset is then biased... Doesn't this

The reviewer is correct that chronologies available on the ITRDB are a fraction of all that

have been created (or published). One can imagine that, if biased one way or another,
the chronologies that are not submitted to the ITRDB might be those with a weaker
climate signal (a form of negative publication bias). Importantly though, our goal in this
study was to reevaluate projections of future growth that have been made in the past
based on chronologies available from the ITRDB, particularly the projections of Williams
et al. (2010, 2013) and Charney et al. (2016). With this goal in mind, our analysis was
based on what is available from the ITRDB.

- A radius of 100 km was applied around the ITRDB site to obtain tree-ring data
from FIA. How robust are results to changes in this radius? I understand that this
radius is a compromise between sample size and representability, and
decreasing the radius would decrease sample size and statistical power, but
given the mountainous habitat of these tree species, a radius of 100 km may
include a huge variation of environmental conditions.

This is a good point raised by the reviewer. In response, we added a new analysis of the
data combined together into a single pool, as requested, eliminating the issue of
selecting a radius around each ITRDB sampling site. The results of that model
corroborate the results from the local-scale comparison: Whether you take the results
from the local-scale comparison (attempting to control for landscape heterogeneity) or
the global comparison that analyses all the data within one single model, we obtain
statistically significant differences between the two databases in both approaches. For a
more detailed description of the model results, we refer the reviewer to lines 174-226
above.

- What was done with overlapping 100-km radii? So, in case ITRDB chronologies were
<200 km apart. This could have led to a re-use of FIA ring-width series, which is clearly
not desirable.

We added a new analysis of the data combined together in a single pool in an LMEM
framework, as suggested. Please see our responses to comments from Reviewer 1
regarding age and database effects on growth variability, in lines 174-226 above.

- Structure of the Supplementary material. This structure is not so clear and not
very logical. Explanation of statistical tests (1.3) precede text on data availability
(1.4) and sampling design (3).

We have changed the order of the supplementary material and divided it in two sections:
First, we introduce the tree-ring collections, followed by a data availability statement and
the description of the climate data. Second, we report the detrending methods applied
and statistical analyses performed.

- It was unclear to me why the climate sensitivity analysis was only performed for
the southwest. This is also not explained in the section 1.4 of the Appendix. (In
that section, reference is made to Williams ref 16, but this appears as 15 in the
main text.)

We focused on those climate parameters because i) the largest differences in standard
deviation were found in the southwestern United States (Fig. 3) and ii) the second focus
of our paper was to investigate the region-wide growth signal and the implication how
FIA and ITRDB datasets differ in their climate response and hence projected growth
trajectory (replicating the methods and region of Williams et al. 2013). We furthermore
offer a new title for the manuscript, to more clearly focus the scope of inference on the
U.S. Southwest.

We thank the reviewer for catching this error, we have now corrected the reference to
Williams et al. (2013).

- I missed advices for users or contributors or the ‘management’ of the ITRDB on
how to avoid, correct or deal with this overestimation.

This is a good question raised by the reviewer and is an issue of ongoing discussion by
dendrochronologists and ecologists. Thought-free use of ITRDB time series for
ecological questions is clearly not advised, especially at the arid edge of the forest
biome (e.g., in the Southwestern US). Williams et al. (2013) and Charney et al. (2016;
and other papers as well) advised caution with respect to the interpretation of results
based on ITRDB data, because they suspected that the data are biased towards
climate-sensitive trees. We have added the following sentence to the main text (lines
322-326) to provide some guidance:

“We are optimistic that the establishment of a new data standard (TriDaS) that allows
additional information about sites (slope, aspect, stand density, etc.) and trees
(diameter, height, pith offset, etc.) to be readily shared will improve the value and
versatility of publicly available data for ecological forecasts of forested ecosystems.”

- I also missed some general notes on whether this overestimation is to be
expected also in other forest types, regions, species.

*A priori* (based on the work of H. C. Fritts), we would expect that the potential for
“oversensitivity” might be greatest in semi-arid, water-limited ecosystems (e.g., the US
southwest and eastern Mediterranean region). Yet, we also have seen that the ITRDB
Douglas-fir samples of the northern Rockies deviate from the representative FIA dataset
– they are much more positively sensitive to summer temperatures – in spite of the fact
that we do not find a clear elevation difference (ITRDB vs. FIA) in that region. So we
really have to be careful in using the ITRDB for large-scale ecological inferences, unless
we know the purpose of the initial sampling, and how well ITRDB sites represent the
entire forest ecosystem.

In response to Reviewer 3’s comment, we have added a paragraph to the discussion to
address this issue (lines 343-326).

- The authors call for more tree-ring data samples in a representative way: this is
evident. But could they also provide more concrete advice on tree-ring sampling
during forest inventories?

Forest inventories are laid out to systematically sample forested areas across a
landscape (state, country) and represent forest structure (including age) and growing
conditions in every forest stand that falls on a forest inventory grid point. A sample
aimed at representing a forest stand would best be carried out by collecting increment
cores in a stratified random way, i.e. sampling one tree per species and size class.

We are in the process of organizing an international workshop focusing on this question
of sampling design, and plan to write a review paper. In the meantime we have added a
citation of Nehrbass-Ahles et al. (2014), which reports on how within-site sampling
design influences estimation of trends in biomass accumulation, and refer to ecological
gradient-based and gridded sampling designs as promising, spatially representative
sampling designs (ms lines 329-333).

- This study considers site selection as the main way in which ‘targeted sampling’ is
conducted. But this is just one of the ways at which dendrochronologists/climatologists
attempt to increase the climate (or common) signal in tree-ring data. Other steps
include: **i)** selection of trees to be sampled (dominant), **ii)** selection of tree-ring series to
be included in analyses; **iii)** selection of series in master chronology; **iv)** selection of
chronologies to be uploaded to the ITRDB (!). What were the effects of the other steps?
**v)** And was the selection of dominant trees in the ITRDB studies done in a similar way
as in the FIA studies?

i) Age effect: ITRDB trees are much, much older than FIA trees, and the LMEM analysis
indicates that age increases climate sensitivity.

ii) The reviewer is correct that “complacent” time series are regularly excluded from
studies focused on forming a site-level chronology with strong climate signal. We don’t
see any way that we could quantify how much of the “oversensitivity” of the ITRDB
sample is due to this step of the process.

iii) Same point as above – there’s no way for us to quantify this aspect of ITRDB
“oversensitivity”.

iv) The reviewer is correct that chronologies available on the ITRDB are a fraction of all
that have been created (or published). One can imagine that, if biased one way or
another, the chronologies that don’t make it onto the ITRDB might be those with weak
climate signal, or those that have been sampled for dendrogeomorphological or -
entomological studies with clear signs of non-climatic disturbance.

v) Trees that are cored on the FIA plots are usually belonging to co-dominant or
dominant canopy classes. Unlike most trees from the ITRDB, that were mostly selected
and cored because they had little competition and no signs of damage – to maximize the
climate signal – sampling at FIA plots does not follow these two criteria.

Reviewers' comments:

Reviewer #2 (Remarks to the Author):

The authors have largely addressed my previous comments, though the age analysis is almost entirely in the supporting material, but they still maintain in the title a level of certitude that is not warranted given the evidence they supply.

Specifically, I have three main issues with the framing.

First, the title is , "Global warming will slow forest growth in the southwestern U.S., but less than previously reported". WILL is strong, given that the analysis still detects a 75% decrease in growth for median future climate for FIA. The title would lead us to believe that it's not that big a deal. Moreover it is statistical and therefore prone to projection outside the range of observations. Shouldn't the authors be more circumspect and state more clearly that the most sensitive fraction of the population (i.e., older, more sensitive trees and those trees nearer the limits of the species' tolerances for water limitation) will likely decrease (not decline) in growth as much as prior estimates have suggested? The title and tone of the paper are not really true to this nuance - the authors do not even discuss the limitations of a statistical approach. Unless I missed it, they also do not cite a highly relevant paper, Restaino et al. 2016 PNAS, which also uses a less biased dataset (at least compared to the ITRDB) for Douglas-fir (one of the species indicated here) and addresses - but does not "predict" - this same issue, i.e., likely decrease in growth. Why would the authors exclude this relevant paper?

Second, the authors uses precipitation and Tmax as predictors despite ample evidence that these are correlative, rather than direct, drivers. Such drivers could be better approximated with other variables such as water balance or possibly vapor pressure deficit. Also, for the late 21st century, it would appear that the extreme low end members for BOTH FIA and ITRDB samples (Figure 5) result in negative RWI. How is that "less than previously reported"? The authors encourage us to brush this away in lines 309-310 - "not to be taken literally". If the model projects something that isn't to be taken literally, why do the authors use such certain language?

Third, related to (1) and (2) above, I think the authors place too much value on an assumed stationary statistical response into possibly no-analog climate space. Specifically, they've projected forward based on decadal seasonal climate the likely responses and assumed that (1) the major climatic limitations and physiological responses will remain constrained to the timing historical predictors)that is, "mean monthly maximum temperatures of the antecedent fall (August to October), current summer precipitation (May to July), and cool-season precipitation (November to March" (lines 150-151). Because tree physiology (like bud set, stomatal response to vapor pressure deficit, and timing of growth) are sensitive to timing of multiple climatic variables across the phenological calendar, the certainty of comparison is unwarranted.

I think either the paper should be rejected in its current state or, alternatively, the authors could much better qualify the limitations of their approach and back off the claim that appears too certain given the limitations of the analysis. Specifically, the authors need caveat the findings clearly and state that their conclusions are based on an analysis that (1) uses climate predictors that are proxies for physiological responses and therefore true growth responses may deviate from the statistical predictions; (2) ignores other variables shown in the literature to be more powerful statistical predictors and that are more closely related to climate-driven physiological responses that affect growth, and (3) assumes seasonality of future climate and tree response to independent variables will remain stationary under possibly no-analog climates.

Reviewer #3 (Remarks to the Author):

Klesse et al conducted a thorough and extensive revision of their initial submission. The revision involved conducting additional analyses, adding and replacing figures and a major changes in text. Overall, the concerns raised by me (Ref 3) based on the initial submission have been dealt with adequately and I am pleased to see that the new statistical analyses that included both types of databases (targeted ITRDB and representative FIA) yielded direct statistical evidence of a database x climate interaction effect. As far as I can judge, the main comment by Reviewer 1 (on the need to conduct pre-whitening of the data) was also adequately addressed: the similarity of Figure 4 for pre-whitened and non-pre-whitened ring-width series is convincing.

The new statistical tests give more certainty and rigour about the effects of sampling on climate sensitivity. I am therefore confident that these new analyses support the authors' claim that sampling method affects climate sensitivity, and that caution is required in interpretation of results from targeted sampling.

I remain somewhat hesitant to extrapolate beyond climatic conditions (temperature) included in the sampled climatic ranges, and to use effects of annual *fluctuations* in temperature on tree growth (the result of the mixed effect models) to make inferences about the effects of future change in the *mean* temperature (the application with CMIP5 models). But I am happy with the way authors call for caution in using the absolute values of the estimated growth decline under future climate.

Overall, I remain very positive about this study, as it provides the much-needed evidence that tree-ring data obtained from studies aiming to reconstruct past climate variability overestimate the climate sensitivity compared to data from representative sampling. This is a very important message, that deserves to be published in a scientific journal with the potential to reach a broad audience.

Reviewer #2 (Remarks to the Author):

The authors have largely addressed my previous comments, though the age analysis is almost entirely in the supporting material, but they still maintain in the title a level of certitude that is not warranted given the evidence they supply.

Specifically, I have three main issues with the framing.

First, the title is , "Global warming will slow forest growth in the southwestern U.S., but less than previously reported". WILL is strong, given that the analysis still detects a 75% decrease in growth for median future climate for FIA. The title would lead us to believe that it's not that big a deal.

We agree with the reviewer that a projected growth decrease of 75% is a lot. At the same time, the reviewer seems here (and below) to be primarily concerned about overstating confidence about the magnitude of the growth decline. Fundamentally, we agree with the reviewer about the caveats associated with projection of a simple statistical model (see below). Because we want to de-emphasize the absolute magnitude of the projected decline in growth, and instead emphasize our certainty about the biased nature of the ITRDB sample compared to the FIA sample, we have made several changes to focus attention on the difference in the growth decrease (of 29%) based on a comparison between the two samples. The first of these changes is to the title: **Projected climate change impacts on forest growth are biased by sampling climate-sensitive trees**. Our intention with this new title is to draw the reader's focus less to the absolute projection and more so to the problem of projection based upon a biased sample. In this context, the purpose of simple statistical projection is to illustrate what implications the biased sampling has.

Moreover it is statistical and therefore prone to projection outside the range of observations.

We agree that extrapolation is the most dangerous part of "ecological forecasting", and have added this as a caveat.

Shouldn't the authors be more circumspect and state more clearly that the most sensitive fraction of the population (i.e., older, more sensitive trees and those trees nearer the limits of the species' tolerances for water limitation) will likely decrease (not decline) in growth as much as prior estimates have suggested?

Indeed, the most sensitive fraction of the population *is* likely to respond as the ITRDB sample would suggest. The point is that this fraction is *very small*. The trees that have been targeted for climate reconstruction are very special, and rare, trees on the landscape. This is well known by practicing dendrochronologists, and is the reason that there has been such a strong emphasis, as new dendrochronologists are trained, on teaching the "site and tree selection" principles. From a carbon cycle science perspective, however, the very small fraction of the population that is highly climate sensitive is almost negligible.

The title and tone of the paper are not really true to this nuance - the authors do not even discuss the limitations of a statistical approach.

We have added several sentences to more clearly address the risks and shortcomings of a (simple) statistical model (see lines 297-319). As above, please note that we have changed the title to de-emphasize the absolute magnitude of the growth projection and give more weight to the difference arising from contrasting samples.

Unless I missed it, they also do not cite a highly relevant paper, Restaino et al. 2016 PNAS, which also uses a less biased dataset (at least compared to the ITRDB) for Douglas-fir (one of the species indicated here) and addresses - but does not "predict" - this same issue, i.e., likely decrease in growth. Why would the authors exclude this relevant paper?

This is an error on our side, thank you for catching it. That publication should actually be cited, but somehow disappeared while merging the reference lists of the main text and the methods (which before were separate). We fixed this mistake.

Second, the authors uses precipitation and Tmax as predictors despite ample evidence that these are correlative, rather than direct, drivers. Such drivers could be better approximated with other variables such as water balance or possibly vapor pressure deficit.

We agree with the reviewer that precipitation and Tmax are correlative predictors of tree ring width variation, and please note the change to the sentence in line 307 to reflect this. While VPD or water balance may be more integrative variables (integrating together aspects of temperature and moisture), we sincerely doubt that they would explain more ring width variation, or substantially more, than precipitation and Tmax. First, VPD and Tmax are very highly correlated in this region (Figure S2 in W2013). Indeed, Williams et al. 2013 showed no real difference in the correlation between the US southwest regional chronology and VPD vs. Tmax ($r=0.91$ vs. 0.88). In this warm and arid region, VPD is strongly dependent on temperature, and as such, our analysis would come to the same conclusion and a very similar difference in projected growth decrease based on the ITRDB vs. FIA samples, if we were to replace Tmax with VPD.

Second, neither VPD nor water balance have been measured at the actual sampling sites; they too, like precipitation and Tmax, would be derived from gridded climate products. Replacing one gridded climate product with another would not change the fact that the ITRDB sample is biased, and in particular, that it is biased at a micro-site scale that is not captured by gridded climate products.

Also, for the late 21st century, it would appear that the extreme low end members for BOTH FIA and ITRDB samples (Figure 5) result in negative RWI. How is that "less than previously reported"? The authors encourage us to brush this away in lines 309-310 - "not to be taken literally". If the model projects something that isn't to be taken literally, why do the authors use such certain language?

We thank the reviewer for encouraging us to reconsider (and clarify) what it is that we are certain about, and what it is that we are not certain about. We have high confidence that the ITRDB sample is biased towards climate-sensitive trees, and that projection of future tree growth based on a biased sample is going to lead to a different value, on average, than a projection based on an unbiased (representative) sample. That is, we are confident that there is an important difference between projections based on the two samples. We are less confident about exactly what that value is – what the growth decline will actually be – for all the reasons pointed out by the reviewer (along with others we included in the submitted version of the manuscript). As noted above, we have changed the title (to: **Projected climate change impacts on forest growth are biased by sampling climate-sensitive trees**) to shift the emphasis towards the contrast in sampling and the fact that it will surely affect <any> projections of future tree growth.

That is, the strong and statistically significant difference in standard deviation between the two datasets, which is detectable independent of detrending method and age, would influence any model of growth projection.

Third, related to (1) and (2) above, I think the authors place too much value on an assumed stationary statistical response into possibly no-analog climate space.

Specifically, they've projected forward based on decadal seasonal climate the likely responses and assumed that (1) the major climatic limitations and physiological responses will remain constrained to the timing historical predictors)that is, "mean monthly maximum temperatures of the antecedent fall (August to October), current summer precipitation (May to July), and cool-season precipitation (November to March" (lines 150-151). Because tree physiology (like bud set, stomatal response to vapor pressure deficit, and timing of growth) are sensitive to timing of multiple climatic variables across the phenological calendar, the certainty of comparison is unwarranted.

We agree with the reviewer that climate-growth responses are unlikely to be stationary as trees experience climatic conditions that are increasingly unusual, and have now highlighted this assumption (see lines 308-310).

To reemphasize what it is that we are certain about: the two chronologies (ITRDB and FIA) are virtually the same ($r=0.93$), the two datasets cover the same region, the **significant difference is in the growth variability of the time series, reflected in the standard deviation of detrended ring widths**, (likely due to differences in soil depth and water capacity, along with age), and as such, any projection of growth based on climate parameters in a (statistical) model will be **with certainty** different between the two samples (ITRDB vs. FIA) – no matter the assumption.

Furthermore, it is extremely unlikely that the climate predictors used (or VPD and water balance) during these broad seasons will become unimportant any time in the foreseeable future – from the timing point of view. Cell division and enlargement (turgor pressure is influenced by soil moisture and atmospheric dryness) will always depend on water recharge during the winter (precipitation) and early growing season dryness (T_{max} , VPD), as well as bud set (depending on resources at the end of the growing season) will always be affected by late summer/fall dryness/hotness (August-October).

I think either the paper should be rejected in its current state or, alternatively, the authors could much better qualify the limitations of their approach and back off the claim that appears too certain given the limitations of the analysis.

As detailed above, we have changed the title and added substantial clarification to acknowledge the assumptions highlighted by the reviewer (see lines 297-319), which we fundamentally agree are limitations of a (simple) statistical model, and are, we note, also true for several of the other papers cited (e.g., Williams et al. 2010, Charney et al. 2016).

Specifically, the authors need caveat the findings clearly and state that their conclusions are based on an analysis that (1) uses climate predictors that are proxies for physiological responses and therefore true growth responses may deviate from the statistical predictions;

Thank you for the suggestion, we have added this phrase to the caveat paragraph (see lines 312-313).

(2) ignores other variables shown in the literature to be more powerful statistical predictors and that are more closely related to climate-driven physiological responses that affect growth, and

Please see our response to this point above.

(3) assumes seasonality of future climate and tree response to independent variables will remain stationary under possibly no-analog climates.

This point also has been added to the discussion (see lines 310-312).

Reviewer #3 (Remarks to the Author):

Klesse et al conducted a thorough and extensive revision of their initial submission. The revision involved conducting additional analyses, adding and replacing figures and a major changes in text. Overall, the concerns raised by me (Ref 3) based on the initial submission have been dealt with adequately and I am pleased to see that the new statistical analyses that included both types of databases (targeted ITRDB and representative FIA) yielded direct statistical evidence of a database x climate interaction effect. As far as I can judge, the main comment by Reviewer 1 (on the need to conduct pre-whitening of the data) was also adequately addressed: the similarity of Figure 4 for pre-whitened and non-pre-whitened ring-width series is convincing.

The new statistical tests give more certainty and rigour about the effects of sampling on climate sensitivity. I am therefore confident that these new analyses support the authors' claim that sampling method affects climate sensitivity, and that caution is required in interpretation of results from targeted sampling.

I remain somewhat hesitant to extrapolate beyond climatic conditions (temperature) included in the sampled climatic ranges, and to use effects of annual *fluctuations* in temperature on tree growth (the result of the mixed effect models) to make inferences about the effects of future change in the *mean* temperature (the application with CMIP5 models). But I am happy with the way authors call for caution in using the absolute values of the estimated growth decline under future climate.

Overall, I remain very positive about this study, as it provides the much-needed evidence that tree-ring data obtained from studies aiming to reconstruct past climate variability overestimate the climate sensitivity compared to data from representative sampling. This is a very important message, that deserves to be published in a scientific journal with the potential to reach a broad audience.

Thank you!

REVIEWERS' COMMENTS:

Reviewer #2 (Remarks to the Author):

I am not entirely satisfied with the authors' rebuttal of my previous review (see below), and when sufficiently caveated, I don't agree that it meets the journal's standard of, "Papers published by the journal represent important advances of significance to specialists within each field." The authors indicate the main conclusion, that the variance is higher in trees collected for climate reconstruction than the population as a whole, has been known for a long time (first discussion paragraph, lines 249-260). This paper puts some numbers on what implications that fact has for the growth of three tree species under future climate, and as such it moves the field forward.

However, I want to point out some problems I see in the rebuttal and revised version.

In the revised version, the magnitude of the difference in population growth response between models calibrated with ITRDB growth fs full samples growth is still the main point. There are enough caveats about things that could change that number that the assertions that the variance difference is the main certainty ring a bit hollow. I suggest the abstract contain a final or penultimate clause suggesting something like, "Although there are large uncertainties associated with future no-analog climate and processes not considered by our statistical approach, simple statistical projection based on either the climate-sensitive ITRDB samples or representative forest inventory samples suggest that projected growth decrease due to climate change is 29% less when based on forest inventory samples"

With respect to VPD and water balance deficit, the authors cite a previous study showing that Tmax and VPD are well correlated, and argue that therefore the assumption that Tmax:Precip interactions are sufficient to approximate the real driving variables is reasonably defended. But then later in the paper, they point out that ITRDB and FIA chronologies are well correlated and that the difference in variance is sufficiently novel that it should be published in Nature Communications. VPD is driven in part by Tmax, but also by relative humidity. Water balance deficit, if calculated appropriately (e.g., Penman Monteith or comparable for potential evapotranspiration) also integrates radiation relationships with canopy, soil water capacity, wind, stomata resistance etc. These are correlated with Tmax, but they are not the same and they can vary differentially in the future. The rebuttal argument makes me think the authors don't fully understand the non-stationarity problem here - model observed and real observed approximate each other, but that's no guarantee that model future and observed future will - because, regardless of sophistication, the statistical model can't know things that will no longer co-vary in the future. Gridded products ARE available for these variables, both for historical and future projections, and the authors would do well to test those as predictors and evaluate the stationarity of both in the historical. Using them also brings up other issues, though. The authors' caveats are marginally sufficient to address this, but it could be clearer. For example, on line 309, you could point out that you only considered temperature and precipitation, and other things have been shown to be important.

The authors never answered my question about simulated negative growth. Instead they reiterated that the main point is that the ITRDB is biased relative to FIA (I quite agree it would be). But if the authors want a secondary point to be that the amount of difference is an indicator of the size of the bias, then the model has to be sound. I get it that all models are wrong but some are useful, but a model that projects negative growth has severe limitations, doesn't it? Maybe I misread something.

Abstract: "We show that U.S. Southwest ITRDB samples overestimate climate sensitivity by 41-59%, because ITRDB trees were sampled at warmer and drier locations, both at the macro-site and micro-site scale, and are systematically older compared to the FIA collection." I think you need a "regional" or "forest" between overestimate and climate sensitivity. The ITRDB samples are

as climate sensitive as they are; they don't overestimate their OWN sensitivity, just the greater populations' sensitivities. Right?

Finally, line 236: It's not a forecast. CMIP5 projections don't have probability distributions associated with them, so you can't quantify the uncertainty probabilistically, right? I think it's properly a projection.

Reviewer #2 (Remarks to the Author):

I am not entirely satisfied with the authors' rebuttal of my previous review (see below), and when sufficiently caveated, I don't agree that it meets the journal's standard of, "Papers published by the journal represent important advances of significance to specialists within each field." The authors indicate the main conclusion, that the variance is higher in trees collected for climate reconstruction than the population as a whole, has been known for a long time (first discussion paragraph, lines 249-260). This paper puts some numbers on what implications that fact has for the growth of three tree species under future climate, and as such it moves the field forward.

We agree with the reviewer that it has been known for at least five decades that relative variation in growth is higher in trees collected for climate reconstruction than the population as a whole; this observation (or fact) lays at the heart of the discipline of dendrochronology (i.e., the site and tree selection principles). We agree also that the important contribution of our paper is the quantification of this bias, and specifically, what implications this has for projections of future tree growth. Publications such as Williams et al. 2010 and Charney et al. 2016 used "known-to-be-biased" (climate sensitive) samples to project future growth changes based on 20th century climate-growth relationships and acknowledged the projections could be an overestimate with respect to biased sampling. With a new, large-scale and representative dataset, we are able to quantify this bias. In the U.S. Southwest, representative forest inventory-based samples are 29% less sensitive to annually varying climate conditions compared to the climate-sensitive ITRDB; this leads to an estimate of how much we should correct projections of future growth based on ITRDB samples compared to forest inventory samples. Below, we show that the ball-park figure of a 29% difference in projected growth decrease does not change notably when replaced with two alternative variables suggested by the reviewer, and thus can be considered a robust estimate of difference (within the context of the kind of simple statistical models used by Williams et al. 2010 and 2013 and ourselves).

However, I want to point out some problems I see in the rebuttal and revised version.

In the revised version, the magnitude of the difference in population growth response between models calibrated with ITRDB growth vs full samples growth is still the main point. There are enough caveats about things that could change that number that the assertions that the variance difference is the main certainty ring a bit hollow. I suggest the abstract contain a final or penultimate clause suggesting something like, "Although there are large uncertainties associated with future no-analog climate and processes not considered by our statistical approach, simple statistical projection based on either the climate-sensitive ITRDB samples or representative forest inventory samples suggest that projected growth decrease due to climate change is 29% less when based on forest inventory samples".

Thank you for the suggestion. We have added this additional caveat to the abstract.

With respect to VPD and water balance deficit, the authors cite a previous study showing that Tmax and VPD are well correlated, and argue that therefore the assumption that Tmax:Precip interactions are sufficient to approximate the real driving variables is reasonably defended. But then later in the paper, they point out that ITRDB and FIA chronologies are well correlated and that the difference in variance is sufficiently novel that it should be published in Nature Communications. VPD is driven in part by Tmax, but also by relative humidity. Water balance deficit, if calculated appropriately (e.g., Penman Monteith or comparable for potential evapotranspiration) also integrates radiation relationships with canopy, soil water capacity, wind, stomata resistance etc. These are correlated with Tmax, but they are not the same and they can vary differentially in the future. The rebuttal argument makes me think the authors don't fully

understand the non-stationarity problem here – model observed and real observed approximate each other, but that's no guarantee that model future and observed future will - because, regardless of sophistication, the statistical model can't know things that will no longer co-vary in the future. Gridded products ARE available for these variables, both for historical and future projections, and the authors would do well to test those as predictors and evaluate the stationarity of both in the historical. Using them also brings up other issues, though. The authors' caveats are marginally sufficient to address this, but it could be clearer. For example, on line 309, you could point out that you only considered temperature and precipitation, and other things have been shown to be important.

To address the reviewer's point, we modeled the regional ring width index time series (i.e., FIA and ITRDB chronologies) as a response to the following four combinations of predictors: tmax+precip, tmax+water balance (WB), vapor pressure deficit (VPD)+precip, and finally, VPD+WB. WB is calculated as precipitation minus potential evapotranspiration, which we derived via the Hargreaves method, and VPD is calculated following Buck 1991. Among these four models, the difference in projected growth change based on FIA vs. ITRDB chronologies falls within 4% of one another (29-33%); these are very similar to the value reported in the (previous version of the) manuscript. Explained variances of these four regressions are all within 5% of one another.

To summarize:

- 1) We find a 31% relative difference comparing the standard deviation of FIA with to the standard deviation of the regional ITRDB chronology.
- 2) The two regional chronologies are strongly correlated with one another ($r=0.93$).
- 3) The climate data for both chronologies are almost the same.
- 4) Using virtually the same predictors, where the responses (i.e. the regional chronologies) differ only in the variance, has the consequence that the regression slopes will always be proportional to each other, no matter which predictor is used and how they co-vary in the future. That means in our case that regardless of the four climate parameter combinations mentioned above, the difference in projected growth change is 29-33%. Which leads us to conclude that the 29% difference between growth projections based on the two samples, using a simple linear model under the assumption of stationarity and linearity, is a robust outcome. Switching out VPD and water balance for temperature and precipitation makes essentially no difference to the main result and its magnitude – that an upwards correction of the projected decrease in tree growth results from representative sampling of trees.

We added a sentence to the methods and the caveat paragraph that we explored other parameters such as climatic water balance and VPD, which did not affect the conclusion of the paper.

We understand that the covariance structure of the various factors that go into the calculation of vapor pressure deficit, and potential and actual evapotranspiration, may differ in the future compared to the past (and present). We have clearly caveated our results with respect to non-stationarity (or non-linearity) of climate growth relationships (see lines 267-274).

The authors never answered my question about simulated negative growth. Instead they reiterated that the main point is that the ITRDB is biased relative to FIA (I quite agree it would be). But if the authors want a secondary point to be that the amount of difference is an indicator of the size of the bias, then the model has to be sound. I get it that all models are wrong but

some are useful, but a model that projects negative growth has severe limitations, doesn't it? Maybe I misread something.

We are gratified that the reviewer is convinced by the evidence we've provided that the ITRDB is biased relative to the FIA collection, and by the statement that "all models are wrong, but some are useful" – this is entirely the spirit in which we view our projections of future tree growth. The fact that our statistical model projects negative growth rates indeed deserves further interpretation/explication. We have added a sentence in lines 244-247 to the effect that the implication is that trees can no longer survive under those conditions. Highly-sensitive trees are sensitive because they are living on the edge. At some point (in time), where growing conditions deviate sufficiently from the climatic niche, trees are likely to go over that edge and die. This is very similar to what is implied by climate envelope models, many of which show large range contractions for needle-leaved evergreens by the end of this century (e.g., Rehfeldt et al. 2006).

Abstract: "We show that U.S. Southwest ITRDB samples overestimate climate sensitivity by 41-59%, because ITRDB trees were sampled at warmer and drier locations, both at the macro-site and micro-site scale, and are systematically older compared to the FIA collection." I think you need a "regional" or "forest" between overestimate and climate sensitivity. The ITRDB samples are as climate sensitive as they are; they don't overestimate their OWN sensitivity, just the greater populations' sensitivities. Right?

Thank you, we have added "regional forest [climate sensitivity]".

Finally, line 236: It's not a forecast. CMIP5 projections don't have probability distributions associated with them, so you can't quantify the uncertainty probabilistically, right? I think it's properly a projection.

Correct, we have changed it.

REVIEWERS' COMMENTS:

Reviewer #2 (Remarks to the Author):

In the revised version, the authors show that their approach is essentially equivalent to what they think I was suggesting. I appreciate their attempt to put this issue to rest.

However, they use a Hargreaves estimation for potential evapotranspiration and fail to respond to the suggestion of using a Penman Monteith, indicating they are either unaware of or are ignoring the difference between the two and therefore agnostic of the point at the center of my critique, which is that you cannot necessarily expect statistical modeling of the interactions between T_{max} and precipitation to adequately capture the ecologically relevant variables more central to the growth limitation in conifers.

The unstated assumption (possibly correct) is that with this much data, any difference is meaningless and the statistical inference that $T_{max}:P$ interaction (or their additive responses) is sufficient and supported. That may well be so, and I appreciate the authors' attempt to clarify with additional analysis. But the authors' assumption that it doesn't matter is partly grounded in the idea that PET and VPD vary with temperature, which is true, but they ALSO vary, in timing and magnitude, with other factors that T_{max} DOES NOT ESTIMATE or correlate with, at least in a simple, linear, no-lag environment, and a Hargreaves estimation does not include those other variables.

I do not think this difference should hinder the potential publication of this paper if the editors deem it appropriate for this journal. However, I am going to explain, hopefully clearly, why I see this response as dodging what's important here.

There is a rich literature on the difference in micrometeorological (e.g., Penman Monteith, Priestly Taylor) vs empirical (Hargreaves, Thornthwaite, others) estimations of potential and actual evapotranspiration. Not all of it is ecologically centric - some is hydrologically centric, but a good start is Yates 1994.

Hargreaves is a function of temperature and radiation, and therefore PET must increase as a function of temperature given day length and no real change in incoming solar radiation. Climatic water deficit that is calculated as PET from Hargreaves and precipitation is of course going to be essentially the same as T_{max} and precipitation. Penman Monteith integrates other variables in an explicit calculation of water limitation by comparing the water available to the plant for respiration to the demand for water from the atmosphere and environment and estimating the deficit NOT as PET-PPT or PPT/PET, but as PET-AET or AET/PET. This approach has the ability to directly consider humidity, storage as snow, soil field capacity, wind, stomatal resistance and other factors that contribute to a complex function. Annual averages or totals based on these methods often correlate, but seasonal differences can be big AND vary with regional hydrology. To quote Yates, "Empirical methods, which are often only temperature based, give significantly different marginal changes to temperature fluctuations when compared with the physically (or micrometeorological) based methods such as the modified Penman equations. Drastically different results are found under the same climate scenarios for a given basin."

The reason this matters is that T_{max} is regionally conserved, and precipitation, while noisier than T_{max} , is also comparatively conserved. Soil field capacity, community aggregated stomatal conductance, radiation, wind, and snow responses (things that affect the AET and thus deficit) vary much more in both space and time, and so the error in growth response to T_{max} and precipitation is variable in space and time, and therefore so is the "bias" of trees selected to be climatically sensitive vs. those that are not. If that error has no spatial or temporal structure (e.g., it is random or IID), it isn't going to matter. If that error has spatial or temporal structure, it DOES matter because the regional comparisons made in this study will not be stable.

It may be that the magnitude effect, given the volume of data available, is large enough this quibble doesn't matter, and the additional analysis presented by the authors attempts to address that. If the authors caveat their conclusions by acknowledging that their choice of analysis method dismisses, rather than refutes, the role of these variables, the analysis and conclusions would be appropriately characterized. The title, then, might need to be stated less strongly - it's true to a point, but it may not be universally true because all these trees are climatically sensitive, just for many of them the climate does not vary much beyond the thresholds that limit their growth.

Reviewer #2 (Remarks to the Author):

In the revised version, the authors show that their approach is essentially equivalent to what they think I was suggesting. I appreciate their attempt to put this issue to rest.

However, they use a Hargreaves estimation for potential evapotranspiration and fail to respond to the suggestion of using a Penman Monteith, indicating they are either unaware of or are ignoring the difference between the two and therefore agnostic of the point at the center of my critique, which is that you cannot necessarily expect statistical modeling of the interactions between Tmax and precipitation to adequately capture the ecologically relevant variables more central to the growth limitation in conifers.

The unstated assumption (possibly correct) is that with this much data, any difference is meaningless and the statistical inference that Tmax:P interaction (or their additive responses) is sufficient and supported. That may well be so, and I appreciate the authors' attempt to clarify with additional analysis. But the authors' assumption that it doesn't matter is partly grounded in the idea that PET and VPD vary with temperature, which is true, but they ALSO vary, in timing and magnitude, with other factors that Tmax DOES NOT ESTIMATE or correlate with, at least in a simple, linear, no-lag environment, and a Hargreaves estimation does not include those other variables.

I do not think this difference should hinder the potential publication of this paper if the editors deem it appropriate for this journal. However, I am going to explain, hopefully clearly, why I see this response as dodging what's important here.

There is a rich literature on the difference in micrometeorological (e.g., Penman Monteith, Priestly Taylor) vs empirical (Hargreaves, Thornthwaite, others) estimations of potential and actual evapotranspiration. Not all of it is ecologically centric - some is hydrologically centric, but a good start is Yates 1994.

Hargreaves is a function of temperature and radiation, and therefore PET must increase as a function of temperature given day length and no real change in incoming solar radiation. Climatic water deficit that is calculated as PET from Hargreaves and precipitation is of course going to be essentially the same as Tmax and precipitation. Penman Monteith integrates other variables in an explicit calculation of water limitation by comparing the water available to the plant for respiration to the demand for water from the atmosphere and environment and estimating the deficit NOT as PET-PPT or PPT/PET, but as PET-AET or AET/PET. This approach has the ability to directly consider humidity, storage as snow, soil field capacity, wind, stomatal resistance and other factors that contribute to a complex function. Annual averages or totals based on these methods often correlate, but seasonal differences can be big AND vary with regional hydrology. To quote Yates, "Empirical methods, which are often only temperature based, give significantly different marginal changes to temperature fluctuations when compared with the physically (or micrometeorological) based methods such as the modified Penman equations. Drastically different results are found under the same climate scenarios for a given basin."

The reason this matters is that Tmax is regionally conserved, and precipitation, while noisier than Tmax,

is also comparatively conserved. Soil field capacity, community aggregated stomatal conductance, radiation, wind, and snow responses (things that affect the AET and thus deficit) vary much more in both space and time, and so the error in growth response to Tmax and precipitation is variable in space and time, and therefore so is the "bias" of trees selected to be climatically sensitive vs. those that are not. If that error has no spatial or temporal structure (e.g., it is random or IID), it isn't going to matter. If that error has spatial or temporal structure, it DOES matter because the regional comparisons made in this study will not be stable.

It may be that the magnitude effect, given the volume of data available, is large enough this quibble doesn't matter, and the additional analysis presented by the authors attempts to address that. If the authors caveat their conclusions by acknowledging that their choice of analysis method dismisses, rather than refutes, the role of these variables, the analysis and conclusions would be appropriately characterized. The title, then, might need to be stated less strongly - it's true to a point, but it may not be universally true because all these trees are climatically sensitive, just for many of them the climate does not vary much beyond the thresholds that limit their growth.

We do understand the difference between Hargreaves (temperature only)-based calculation of climatic water deficit vs. Penman-Monteith type methods and we agree with the reviewer that the additional factors mentioned (e.g., soil field capacity, humidity, wind, community-aggregated stomatal conductance, storage as snow) are influential. Our response, below, is four-fold.

First, we understand the reviewer's point that the temperature and precipitation data extracted from ClimateNA, used as predictors in our statistical model, do not perfectly capture the climatic water deficit conditions experienced by trees on the ground – this is a scaling problem common to global change studies. Unfortunately, ITRDB coordinates predating the introduction of GPS have a precision of two digits after the decimal point. Thus, we have poor information on the true location of those sampling sites. In addition, both newer and older chronologies lack metadata on aspect and slope, since the ITRDB was not designed to store this information. Landscape position strongly influences net radiation, runoff, and therefore evapotranspiration and ecohydrology. Under these circumstances, we don't see how we could faithfully use a Penman Monteith method to calculate actual evapotranspiration. As Hargreaves and Allen (2003) wrote: "Where data quality is questionable [...] the 1985 Hargreaves method [is] recommended." **We have added a sentence to the last paragraph of the discussion to highlight the fact that additional metadata (aspect, slope, soil depth) would enable closing of the scaling gap between gridded climate products and the fine-scale conditions actually influencing individual tree growth, and hence improved estimation of climate sensitivities (see lines 330-333).**

A second point is that we do not agree that the "Penman Monteith" variables mentioned above are "*more central to the growth limitation in conifers*" than precipitation or temperature – a century of research in dendrochronology and dendroclimatology shows that cool season precipitation and warm season temperature are important variables influencing conifer growth in the U.S. Southwest (see Fritts 1976, Adams and Kolb 2005, Williams et al. 2010, and Williams et al. 2013, which are citations 5, 13, 14, and 16 in the manuscript).

Third, we fully agree that the variables highlighted by the reviewer are important in *modulating* the response of conifer growth to precipitation and temperature. We have hypothesized, in fact, that

part of the difference between the ITRDB and FIA samples is attributable to micro-site conditions, which include the micrometeorological/ecohydrological factors raised by the reviewer – lower soil field capacity on steep, rocky slopes (which we already emphasized in our discussion of the “microsite selection bias”, see lines 263-265), along with possibly higher wind and lower humidity on exposed ridges. In other words, rather than invalidating our results or conclusions, the reviewer’s point dovetails with our own interpretation of **why** the ITRDB vs. FIA samples behave differently. **We have added a new sentence in the discussion (where we had previously pointed out this “micro-site selection bias”) to further highlight the importance of the micrometeorological/ecohydrological variables emphasized by the reviewer (see lines 265-269).**

Fourth, the reviewer focuses on spatial structure of the “*the error in growth response [...] of trees selected to be climatically sensitive vs those that are not*”. We have read and reread (and reread...) this paragraph of the review and are struggling to understand exactly what the reviewers’ point is. So, we will address several possibilities that we can imagine. If the reviewer’s point is that the ITRDB vs. FIA samples could have different spatial patterns of error, we want to point out that the climate predictors used to infer climate sensitivity for the ITRDB and FIA datasets are nearly identical ($r=0.98$ for t_{max} , and $r=0.97$ for precipitation), because the extent (the original grid points of the $0.5^\circ \times 0.5^\circ$ -resolved CRU) covers almost exactly the same area. Any coarse-scale calculation of drought indices like water balance (independent of PET calculation method) cannot have a spatial bias for this reason, and fine-scale climate conditions cannot be calculated because of a lack of metadata (e.g., landscape position) as above.

Another possible interpretation of the reviewer’s comment is that the ITRDB trees may “appear” more climate-sensitive than they are because what they are experiencing on the ground is more extreme (in terms of actual evapotranspiration) than what the climate data suggest (because of the missing micrometeorological/ecohydrological/microsite information). We fully agree that this is (probably) true, consistent with the hypothesized microsite selection bias laid out in the manuscript (as above), but this highlights again the **unrepresentative nature of the ITRDB sample** - ITRDB trees were selected for the microsite conditions that make them more climate-limited and climate-sensitive than the average tree. Indeed, the point of our paper is the comparison between that (ITRDB) sample – a biased and unrepresentative sample of trees that are more climate-limited than average, more climate-sensitive than average – vs. a sample that comes as close to unbiased and representative as we have available, or are likely to have available any time in the near future (the FIA sample), by virtue of its forest inventory nature. A forest inventory plot network is explicitly designed to be representative. The plots land where they land. If the reviewer wishes to make the point that ITRDB trees are a spatially biased sample of the trees on the landscape, we couldn’t agree more. If the reviewer wishes to point out that precipitation and temperature alone do not explain the full picture of growth response, we agree as well (as above). But neither point challenges the validity of the comparison that is central to our analysis – between a biased and unrepresentative sample vs. an unbiased and representative sample (again, as nearly so as is available at this time). We have already added (in a previous revision) a sentence to the discussion that acknowledges – to paraphrase – that future growth responses to climate are likely to deviate from the predictions of our statistical model because of the influence of additional plant-relevant environmental variables, and how their variance-covariance structure may change in the future (lines 292-296).

In other words, we're not sure what more this reviewer wants. We stand by the fundamentals of our analysis: that precipitation and temperature are ecologically-relevant climate variables influencing the growth of conifers (in general) and specifically conifers in the US Southwest; that some trees are more sensitive to climate than others, particularly trees living at the edge of a species' or population's climatic tolerance; that the ITRDB sample is not representative, and is biased towards climate-limited, climate-sensitive trees; and finally, that this has consequences for projections of future tree growth under future climate – a forest inventory-based tree-ring network shows a weaker decline in growth compared to the climate-limited, climate-sensitive sample of the ITRDB.